# Addressing challenges in speaker anonymization to maintain utility while ensuring privacy of pathological speech

Soroosh Tayebi Arasteh [1,2,3] ✉, Tomás Arias-Vergara[1], Paula Andrea Pérez-Toro[1], Tobias Weise[1,2], Kai Packhäuser [1], Maria Schuster[4], Elmar Noeth [1], Andreas Maier [1] & Seung Hee Yang[2]

## Abstract

**Background** Integration of speech into healthcare has intensified privacy concerns due to its potential as a non-invasive biomarker containing individual biometric information. In response, speaker anonymization aims to conceal personally identifiable information while retaining crucial linguistic content. However, the application of anonymization techniques to pathological speech, a critical area where privacy is especially vital, has not been extensively examined. **Methods** This study investigates anonymization's impact on pathological speech across over 2700 speakers from multiple German institutions, focusing on privacy, pathological utility, and demographic fairness. We explore both deep-learning-based and signal processing-based anonymization methods. **Results** We document substantial privacy improvements across disorders—evidenced by equal error rate increases up to 1933%, with minimal overall impact on utility. Specific disorders such as Dysarthria, Dysphonia, and Cleft Lip and Palate experience minimal utility changes, while Dysglossia shows slight improvements. Our findings underscore that the impact of anonymization varies substantially across different disorders. This necessitates disorder-specific anonymization strategies to optimally balance privacy with diagnostic utility. Additionally, our fairness analysis reveals consistent anonymization effects across most of the demographics. **Conclusions** This study demonstrates the effectiveness of anonymization in pathological speech for enhancing privacy, while also highlighting the importance of customized and disorder-specific approaches to account for inversion attacks.

## Plain Language Summary

When someone's way of speaking is disrupted due to health issues, making it hard for them to communicate clearly, it is described as pathological speech. Our study explores whether this type of speech can be modified to protect patient privacy without losing its ability to help diagnose health conditions. We evaluated automatic anonymization for over 2,700 speakers. The results show that these methods can substantially enhance privacy while still maintaining the usefulness of speech in medical diagnostics. This means we can keep speech data private whilst still being able to use it to identify health issues. However, our results show the effectiveness of these methods can vary depending on the specific condition being diagnosed. Our study provides a method that can help maintain patient privacy, whilst highlighting that further customized approaches will be required to ensure optimal privacy.

The advent of speech as a pivotal component in digital technology has accentuated privacy concerns due to the inherent biometric information speech carries[1–3], particularly highlighted in its role as a biomarker extensively utilized in healthcare applications[4], such as Parkinson's[5,6] and Alzheimer's[7] diseases detection or speech therapy[8], for its cost-effectiveness and non-invasive nature. However, the advent of deep learning necessitates an ever-increasing volume of speech data for algorithm training. Despite the daily influx of patients with speech or voice disorders at various institutions, leveraging this data for research is hampered by stringent privacy regulations. As speech data can reveal a plethora of personal information, there is

an urgent need for privacy-preserving technologies in voice data usage. Consequently, the scope of data available for public research use remains narrowly limited. Addressing this issue, speaker anonymization[9,10] emerges as a pivotal strategy, aiming to obscure personally identifiable information while preserving essential linguistic and speech characteristics[11,12]. This approach is particularly pertinent in the healthcare sector, where the accuracy and reproducibility of speech biomarkers are paramount for advancing medical diagnostics and treatments[13]. Therefore, finding ways to expand the pool of publicly available training data without breaching privacy norms is crucial for the progression of medical speech technology applications.

[1]Pattern Recognition Lab, Friedrich-Alexander-Universität Erlangen-Nürnberg, Erlangen, Germany. [2]Speech & Language Processing Lab, Friedrich-Alexander-Universität Erlangen-Nürnberg, Erlangen, Germany. [3]Department of Diagnostic and Interventional Radiology, University Hospital RWTH Aachen, Aachen, Germany. [4]Department of Otorhinolaryngology, Head and Neck Surgery, Ludwig-Maximilians-Universität München, Munich, Germany. ✉e-mail: soroosh.arasteh@fau.de

Privacy-preserving data processing has seen substantial growth, motivated by an increasing need for privacy protection. The VoicePrivacy 2020 and 2022 challenges[10,14,15] have been pivotal, setting a framework for defining and exploring speaker anonymization as an essential element of voice biosignal. These initiatives have led to innovations in automatic anonymization methods, including deep-learning-based (DL-based) techniques, such as the extraction and replacement of speaker identity features (x-vectors)[16,17], and signal modification methods, like anonymization using the McAdams coefficient[18,19]. Such endeavors have spurred advancements in anonymization technologies. For example, Mawalim et al.[20] illustrated that phase vocoder-based time-scale modification with pitch shifting offers superior anonymization for healthy speech without sacrificing utility. Khamsehashari et al.[21] developed a voice conversion approach utilizing Emphasized Channel Attention, Propagation, and Aggregation in a time delay neural network (ECAPA-TDNN) to embed speaker identities more effectively. Moreover, Meyer et al.[22] highlighted the successful application of Generative Adversarial Networks in speaker anonymization, while Perero-Codosero et al.[23] utilized autoencoders for this purpose. Furthermore, Srivastava et al.[24] delved into design choices for pseudo-speaker selection and, in another study[25], analyzed the privacy-utility tradeoffs in x-vector-based methods[17].

Despite substantial advances, current research demonstrates a notable gap in the study of anonymization methods tailored to pathological speech, where patient privacy concerns are paramount. Tayebi Arasteh et al.[13] have recently pointed out that the unique attributes of pathological speech make it more readily identifiable than its healthy counterpart, raising vital questions about the privacy-utility balance in its anonymization and the treatment of its biomarkers. While some research, such as the work by Hernandez et al.[26], has delved into articulation, prosody, phonation, and phonology features in anonymized dysarthric speech to differentiate between healthy and pathological speech, and Zhu et al.[27] have assessed anonymization's impact on speech-based diagnostics for COVID-19, these efforts remain limited. They typically concentrate on specific speech or voice disorders, depend on small datasets, or consider pathological speech unrelated to speech or voice disorders, highlighting the need for more comprehensive analyses in this domain.

In response, our study conducts a comprehensive analysis of the impact of anonymization on pathological speech biomarkers, utilizing a large-scale dataset of over 2700 speakers from various institutions. This dataset includes a wide array of disorders such as Dysarthria[28] (a motor speech disorder affecting muscle control), Dysglossia[29] (a condition affecting speech by changes of the articulatory organs, e.g., due to oral cancer), Dysphonia[30] (voice disorders affecting vocal fold vibration), and Cleft Lip and Palate (CLP)[31–34] (a congenital split in the upper lip and roof of the mouth), thus providing a broad basis for generalizable insights into pathological speech anonymization. Additionally, we meticulously explore the balance between privacy enhancement and the utility of pathological speech data, including an examination of demographic fairness implications.

This study aims to investigate whether anonymization modifies the diagnostic markers within pathological speech while balancing privacy-utility and privacy-fairness considerations, suggesting the potential for applying automatic anonymization to pathological speech. Our findings reveal that although anonymization alters the diagnostic markers within pathological speech, it achieves advantageous balances in privacy-utility and privacy-fairness, thus underscoring the feasibility of employing automatic anonymization for pathological speech. Moreover, our analysis identifies substantial variability in the anonymization effects across different disorders, showcasing a complex interplay between anonymization and the specifics of pathological conditions.

## Methods
### Speech dataset
The dataset used in our research comprised a wide array of speech utterances from across Germany. It featured a median participant age of 17, with a mean age of 30 years (± 25 years standard deviation), covering ages from 3 to 95 years. Table 1 offers an overview of the dataset demographics, including voice and speech disorder distributions, and gender breakdown.

**Table 1 | Dataset characteristics**

| Subset | | Total Speakers (Female/Male) [n] (%) | Total Utterances (Female/Male) [n] (%) | Total Duration (Female/Male) [hours] (%) | Average Age (Female/Male) [mean ± std] | Average WRR (Female/Male) [mean ± std] |
|---|---|---|---|---|---|---|
| Overall | | 2742 (47%/53%) | 101,209 (50%/50%) | 191.05 (47%/53%) | 30.14 ± 25.21 (26.57 ± 23.99/ 33.30 ± 25.83) | 61.48 ± 15.97 (65.49 ± 14.87/ 57.94 ± 16.07) |
| Adults | Overall | 1056 (37%/63%) | 45,144 (47%/53%) | 82.87 (45%/55%) | 58.71 ± 17.20 (58.71 ± 19.48/ 58.71 ± 15.73) | 62.27 ± 16.50 (68.48 ± 14.35/ 58.64 ± 16.61) |
| | Dysarthria | 355 (54%/46%) | 8456 (43%/57%) | 15.53 (43%/57%) | 62.98 ± 17.13 (62.46 ± 17.83/ 63.59 ± 16.31) | 69.82 ± 12.30 (71.84 ± 10.83/ 67.43 ± 13.49) |
| | Dysglossia | 542 (28%/72%) | 34,772 (49%/51%) | 63.60 (46%/54%) | 61.04 ± 11.61 (62.57 ± 14.26/ 60.45 ± 10.37) | 56.95 ± 16.11 (62.22 ± 16.09/ 54.91 ± 15.67) |
| | Dysphonia | 78 (10%/90%) | 909 (9%/91%) | 1.83 (7%/93%) | 59.13 ± 9.92 (52.64 ± 12.31/ 59.87 ± 9.44) | 52.07 ± 15.92 (56.48 ± 22.32/ 51.56 ± 15.15) |
| | Control | 81 (48%/52%) | 1007 (50%/50%) | 1.91 (44%/56%) | 23.98 ± 16.04 (26.52 ± 15.94/ 21.62 ± 15.96) | 74.70 ± 14.84 (78.63 ± 7.38/ 71.05 ± 18.73) |
| Children | Overall | 1,734 (53%/47%) | 56,065 (52%/48%) | 108.18 (49%/51%) | 12.28 ± 4.00 (12.61 ± 4.03/ 11.90 ± 3.95) | 61.36 ± 15.82 (64.50 ± 14.96/ 57.84 ± 16.04) |
| | CLP | 468 (45%/55%) | 17,008 (43%/57%) | 38.10 (40%/60%) | 9.80 ± 4.22 (9.78 ± 4.25/ 9.82 ± 4.20) | 48.67 ± 17.24 (50.67 ± 17.86/ 47.06 ± 16.58) |
| | Control | 1,266 (56%/44%) | 39,057 (56%/44%) | 70.08 (54%/46%) | 13.19 ± 3.51 (13.45 ± 3.55/ 12.86 ± 3.42) | 66.05 ± 12.32 (68.59 ± 11.06/ 62.83 ± 13.06) |

This table summarizes participant counts, gender distribution, utterance totals, hours of speech, age groups, and word recognition rates (WRRs), presented as mean ± standard deviation (std). It categorizes participants into adults (21 years and above) and children (20 years and under), including subgroups for healthy controls, Dysglossia patients, Dysarthria and Dysphonia cases, and children with Cleft Lip and Palate (CLP).

**Data collection.** Data were collected from 2006 to 2019 during regular outpatient examinations at the University Hospital Erlangen and at over 20 different locations across Germany for the recording of control speakers. Every patient during a specialized consultation was invited to participate in the study. The study and the methods were performed in accordance with relevant guidelines and regulations and approved by the University Hospital Erlangen's institutional review board with application number 3473 and respected the Declaration of Helsinki. Informed consent was obtained from all adult participants as well as from parents or legal guardians of the children. Patients and control speakers were informed about the study's procedure and goals before consenting to participate and providing informed consent.

Recordings were made using a standardized procedure which included consistent settings, microphone setups, and speech tasks. Non-native speakers and patients whose speech was substantially disturbed by factors other than the targeted disorders were excluded. The Program for Evaluation and Analysis of all Kinds of Speech disorders (PEAKS)[35], an open-source tool widely used in the German-speaking scientific community, was employed to document and manage the database. Recordings were captured at a 16 kHz sampling frequency and a 16-bit resolution, featuring subjects who are native German speakers, including various local dialects.

**Speech features.** The dataset included different causes with their main or prominent features of pathological speech, e.g., Dysphonia, refers to voice disorder containing phonation features, Dysglossia refers to articulation disorders containing mostly phonetic and sometimes phonation features, Dysarthria refers to speech disorder containing phonation, phonetic and prosody features, and CLP refers to speech and resonance disorder containing phonetic features, hyper- and hyponasality, and sometimes phonation features. Supplementary Table 1 provides an overview of the expected features for different clinical groups.

**Exclusion criteria.** The cohort employed in our study represents a meticulously curated subset of the dataset described in ref. 13, where it is delineated that our initial collection consisted of 216.88 h of recordings from $n = 4121$ subjects. To refine this dataset to a clean and unbiased selection, we adhered to all exclusion criteria mentioned in ref. 13, which encompassed data cleaning, ensuring speech quality and noise standards, and the elimination of multi-speaker utterances. Additional steps undertaken in this study include: (i) Acknowledging the distinct speech characteristics between adults and children[13], we categorized the dataset into two primary subsets. Adults, defined as individuals over 20 years of age, were tasked with reading Der Nordwind und die Sonne, a phonetically rich German adaptation of Aesop's fable The North Wind and the Sun[35]. This text comprises 108 words, 71 of which are unique. Conversely, children participated in the Psycholinguistische Analyse kindlicher Sprechstörungen (PLAKSS)[36] test, which involved naming pictograms across slides, covering all German phonemes in various positions. Given the tendency of some children to describe pictograms with multiple words, and the occasional extra words between target words, recordings were automatically segmented at pauses exceeding 1s[35]. (ii) Adults' subset focused on utterances characterized by Dysarthria[28], Dysglossia[29], and Dysphonia[30], alongside healthy control samples. Utterances with ambiguous or mixed pathologies or those representing conditions with a scant number of data points were excluded. (iii) For children, the emphasis was placed on utterances from individuals with CLP conditions — the most prevalent cranial malformation characterized by an incomplete closure of the vocal tract[31,32,34] — as well as from healthy controls.

**Experimental design**
Given the interdisciplinary nature of our study (Fig. 1), which incorporates a variety of criteria, establishing clear metrics, criteria, and statistical methods is crucial for foundational clarity. We detail our approaches to evaluating the effectiveness of anonymization, as well as assessing the utility and fairness of pathological speech data. Subsequently, we describe the anonymization techniques we utilized. Then, a breakdown of all different experiments performed in this study is given.

**Evaluation criteria**
**Anonymization measure (privacy).** To evaluate the effectiveness of anonymization, we employ automatic speaker verification (ASV) techniques, aligning with established standards in the field, such as those set by the VoicePrivacy 2020 and 2022 challenges[10,14,15]. Utilizing a pre-trained module, optimized on the LibriSpeech[37] dataset, we fine-tune this system to recognize specific speakers. This involves exposing the module to random utterances, which may belong to the target speaker or an imposter from the dataset. The goal is to achieve an equal error rate (EER) where the rate of false acceptance (FA) matches that of false rejection (FR), indicating the system's difficulty in distinguishing the speaker's identity post-anonymization. We adopt the EER as our primary metric for assessing anonymization[38], a critical metric in ASV[39]. It inversely correlates with verification ease—a lower EER indicates a more straightforward speaker verification process[40]. This metric aims to find a threshold where the false acceptance rate (FAR) and false rejection rate (FRR) are balanced, using cosine distance scores for similarity measurement. An increase in EER post-anonymization, relative to the baseline EER, signifies enhanced anonymization efficacy. EER values are reported as percentages throughout this paper.

**EER calculation process.** The data preprocessing followed established protocols[13,41–43], starting with the removal of intervals with sound pressures below 30 dB. Voice activity detection[44] was then applied to eliminate silent segments from the utterances, utilizing a 30 ms window length, a maximum silence threshold of 6 ms, and an 8 ms moving average window. The final feature set comprised 40-dimensional log-Mel-spectrograms, calculated with a 25 ms window length and 10 ms steps, employing a short time Fourier transform (STFT) of size 512.

A pretrained ASV network on the LibriSpeech[37] dataset (360-clean subset), was employed. This network comprised three long short-term memory (LSTM)[45] layers with 768 hidden nodes, followed by a linear projection layer, and was trained using the Generalized End-to-End (GE2E)[42] loss with the Adam[46] optimizer. For an in-depth discussion of the ASV network's training and evaluation, readers are directed to refs. 13,41,42, as these details extend beyond the scope of our current study.

The EER calculation process involved comparing pairs of positive (same speaker) and negative (different speakers) utterances to verify speaker identity. Initial EER values for various dataset subsets were calculated and then compared to their anonymized counterparts to assess the impact of anonymization on speaker verifiability.

**Biomarker analysis measure (utility).** Pathological speech is characterized by a range of subjective and objective metrics, including articulation[47], prosody[48], and phonation[26,49], among others. Recognizing the complexity and diverse nature of these biomarkers[3,13], our study opted for a deep learning-based approach to biomarker assessment. This method does not focus on extracting specific features but allows the network to autonomously identify pertinent features distinguishing healthy speech from various voice and speech disorders. This approach facilitates the application of standard classification metrics such as area under the receiver operating characteristic curve (AUROC), accuracy, sensitivity, and specificity in evaluating the utility of pathological speech biomarkers and standard statistical analyses.

**Classification process.** During data preprocessing, if present, drifting noise was removed by applying a forward-backward filter[50]. The final feature set comprised 80-dimensional log-Mel-spectrograms, employing an STFT of size 1024.

Due to the two-dimensional nature of log-Mel-spectrograms, we leveraged state-of-the-art pretrained convolutional networks designed for image classification to maximize feature extraction accuracy[51,52]. We

**Fig. 1 | Illustration of pathological speech data analysis challenges and solutions. a** This panel depicts a scenario where a patient, referred to as Bob, visits a clinic for speech therapy. His recorded speech is conventionally de-identified (by solely omitting Bob's name and personal details from the folder and filenames) before storage for further analysis and therapy use. However, a data breach occurs, allowing an unauthorized listener to access and identify Bob from his speech data due to the lack of speech anonymization. **b** In contrast, this scenario introduces an additional step of automatic anonymization before Bob's speech data is stored in the clinic. This process effectively obfuscates Bob's unique speaker characteristics, preventing potential eavesdroppers from recognizing him through his speech data. Importantly, despite the anonymization, the modified speech data retains sufficient integrity for pathological assessment by clinicians.

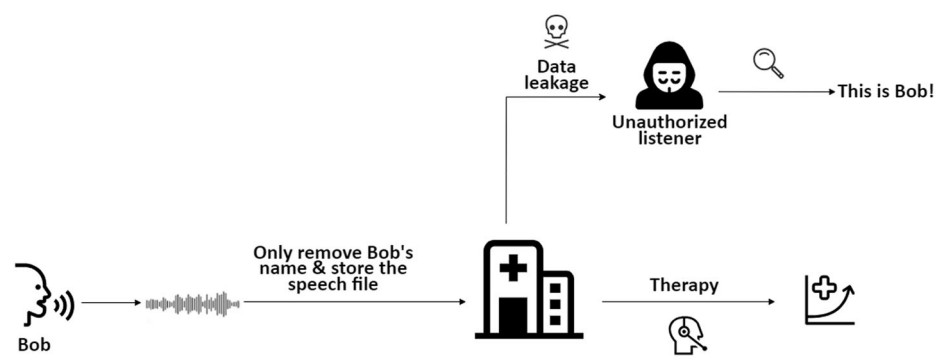

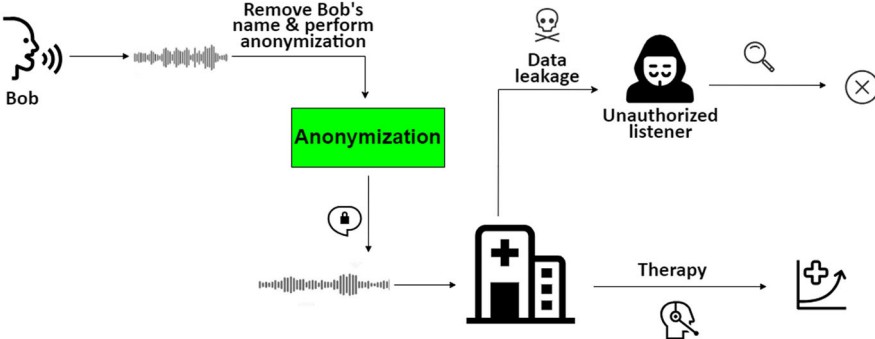

specifically selected a ResNet34[53] model pretrained on the large-scale ImageNet[54] dataset, which contains over 14 million images across 1000 categories, adhering to the architecture proposed by He et al.[53], featuring a $(7 \times 7)$ convolution first layer producing an output with 64 channels, and a final linear layer reducing the $(512 \times 1)$ output feature vectors to 2. The sigmoid function was applied to transform output predictions into probabilities, totaling around 21 million trainable parameters in the network.

We employed a batch size of 8, selecting 8 utterances per speaker randomly for each batch. Given the varying lengths of log-Mel-filterbank energies, 180 frames (about 3 s) were chosen at random for inclusion in training. The network inputs were sized $(8 \times 3 \times 80 \times 180)$, reflecting the batch size, channel size (adjusted to 3 to match the pretrained network's expectation, with log-Mel-spectrograms replicated three times), log-Mel-spectrograms size, and frame size. Training spanned 150 epochs, optimizing with the Adam[46] optimizer and a learning rate of $5 \times 10^{-5}$, utilizing binary weighted cross-entropy for loss calculation.

**Statistics and reproducibility**
Statistical analysis was conducted using Python v3.9, leveraging the SciPy and NumPy libraries. For each disorder subset and specific experiment, speakers were randomly allocated to training (70%) and test (30%) groups. This random allocation was consistent across experiments to ensure that the same training and test subsets were used for comparing anonymized data with original data, facilitating paired analyses to account for random variations. The division aimed to prevent overlap between training and test data. To address potential imbalances in the dataset, particularly where there was a limited number of healthy controls (81 in the adult subset), we adjusted the patient-to-control ratio. In cases of Dysarthria and Dysglossia

with ample patient data ($n = 355$ and $n = 542$, respectively), we capped patient speakers at twice the number of controls. In the children's subset, which had more controls, we sampled controls up to 1.5 times the number of patients to maintain balance. The composition of the final training and test sets, ensuring a fair comparison between the two anonymization methods, was as follows: Training sets comprised $n = 168$ speakers (Dysarthria detection), $n = 168$ (Dysglossia detection), $n = 110$ (Dysphonia detection), and $n = 887$ (CLP detection). Corresponding test sets included $n = 73$ (Dysarthria detection), $n = 73$ (Dysglossia detection), $n = 49$ (Dysphonia detection), and $n = 381$ (CLP detection). AUROC was selected as the primary evaluation metric, with accuracy, sensitivity, and specificity serving as secondary metrics. Considering each speaker contributed multiple utterances, and to account for the random sampling of utterances in training and testing, each test phase was repeated 50 times to reduce potential random biases, with evaluations strictly paired for a consistent comparison between anonymized and non-anonymized data. Results are expressed as mean ± standard deviation. Statistical significance was determined using a two-tailed unpaired t-test, setting a significance level at $p < 0.05$. Pearson's correlation coefficient was utilized to measure the correlation between EER and AUROC (i.e., the privacy-utility tradeoff).

**Anonymization method**
Anonymization techniques usually fall into two primary categories: (i) DL-based synthesization methods and (ii) signal-level modification methods. To maintain generality, we considered both categories in our study.

**DL-based synthesization method.** These types of methods initiate by converting the waveform into the frequency domain and segregating the

speaker identity features from other acoustic features. Subsequently, specific modifications are applied solely to the speaker identity features, ensuring no other feature is altered. The modified frequency domain features, typically represented as Mel-spectrograms, are then re-synthesized back into the time domain using a synthesizer known as a vocoder in the text-to-speech[55] context. This approach is termed DL-based because critical components, such as the vocoder and speaker identity extractor, are trained using DL-based methods.

The most prevalent applications of these methods are integrated with voice conversion[56] algorithms, where after isolating speaker identity features, they are substituted with those of another speaker. This process effectively changes one's voice to mimic another's. However, for anonymization purposes, especially in scenarios involving over 2700 speakers as in our study, mapping each speaker to a specific target is impractical. Therefore, we opt for a simplified approach, focusing on altering only the pitch frequency of the speakers before re-synthesizing the signal with a vocoder.

The same data preprocessing procedure as for the classification network was utilized.

To prevent reconstruction of the original signals, the pitch shift magnitude was chosen at random for different speakers. However, these Modifications were carefully applied to ensure audibility, gender preservation, minimal age alteration, and naturalness. Supplementary Fig. 1 details the proposed pitch shift magnitude, noise addition, and denoising processes.

To re-synthesize the speech waveforms from the modified Mel-spectrograms, the HiFi-GAN[57] vocoder was utilized, a state-of-the-art voice synthesizer pre-trained on the LibriTTS[58] corpus, a diverse corpus of healthy English speech encompassing 585 h of audio. This approach ensured that the resulting speech maintained high fidelity and naturalness.

**Signal-level modification method.** Signal-level anonymization techniques utilize signal processing to modify speech signals without the need for model training. A prominent method involves the McAdams coefficient, which alters speaker characteristics by adjusting the spectral formant positions via linear predictive coding analysis. The technique involves analyzing speech frame by frame to extract features, then adjusting the spectral positions based on the McAdams coefficient[19]. This adjustment changes the speaker's perceived identity by modifying the phase of certain frequencies, while leaving others untouched. The method is effective for standard speech samples, targeting key spectral features for anonymity. The speech is then reconstructed with adjusted features, ensuring both the anonymization of the speaker and the preservation of speech clarity. Our approach adopts and refines a variation of the anonymization method by Patino et al.[18], initially introduced in the VoicePrivacy 2022 Challenge[15]. This adaptation is particularly advantageous for our application because it eliminates the necessity for mapping original speakers to target ones. A key aspect of this method is the utilization of the McAdams coefficient as an anonymization metric; in a specific range, a higher coefficient indicates a lower degree of anonymization, allowing for a customizable balance between speaker anonymity and speech intelligibility.

**Experiments breakdown**

The breakdown of the experiments performed in this study are detailed below.

**Performance evaluation of anonymization methods for pathological speech.** The DL-based method is implemented as per Supplementary Fig. 1. For the McAdams coefficient method, given our speech dataset's 16 kHz sampling rate, we opt for dynamic selection of the McAdams coefficient, randomly choosing a value between 0.75 and 0.90. This approach introduces additional randomness and complicates potential data reconstruction. We noted that coefficients above 0.90 minimally affect anonymization, aligning with the VoicePrivacy 2022 Challenge baseline, while those below 0.75 begin to degrade audio quality and naturalness.

**Exploring synthesizing effects.** Notably, nasality changes are often observed in the deeper frequencies, which could be particularly relevant in disorders like CLP, where pitch shifts might substantially alter pathological features and formant information. To assess the general impact of DL-based anonymization methods, we delve into a simplified scenario: speech signals are converted into log-Mel-spectrograms yet undergo no alterations prior to being synthesized through the HiFi-GAN vocoder. In an ideal setting, where the vocoder performs impeccably, this approach should theoretically leave both utility and privacy unaffected. Although this experiment does not directly contribute to anonymization, it provides valuable insights into how these processes might influence the pathological utility of speech data. This evaluation aids in distinguishing the appropriateness of DL-based versus signal-level based methods for the anonymization of pathological speech.

**Privacy-utility tradeoff.** We then examined the effects of varying privacy levels. Using the McAdams coefficient method, we systematically adjusted the coefficient from 0.5 to 1.0 (in increments of 0.1) and train separate classification models for each. This allowed for an in-depth analysis of the privacy-utility tradeoff in pathological speech and helps understand their correlation.

**Privacy-fairness tradeoff.** We evaluated the balance between privacy and fairness by analyzing demographic subgroups within our dataset. A fair classification network, in this context, is defined as one that maintains equal performance in detecting speech or voice disorders across all patient subgroups, both before and after anonymization[59]. To assess this, we not only compared AUROC performance and EER privacy metrics across different subgroups but also employed statistical parity difference (PtD)[60] as a measure of demographic fairness. This metric represents the accuracy disparity between minority and majority classes, with ideal values being zero—indicating no discrimination. Positive values suggest a benefit to the minority class, whereas negative values indicate potential bias against these groups[61]. The demographic subgroups analyzed included gender (female and male) and age (adult and child), aiming to ensure equitable performance across these variables.

**Anonymization in diverse scenarios.** Acknowledging Tayebi Arasteh et al.[13]'s findings that diversity in speakers and disorders substantially enhances the performance of ASV systems, making anonymization more challenging, we consolidated all patient data into a general patient set and all control data into a general control set. We undertook a task of detecting pathological speech across this combined dataset to evaluate the effectiveness of anonymization methods for large-scale pathological speech corpora. This comparison between original and anonymized speech data aimed to determine the feasibility of applying automatic anonymization to pathological speech in extensive datasets.

**Broadening utility assessment.** To ensure the robustness of our findings and mitigate task-specific biases, we undertook a multiclass classification challenge. Moving beyond the binary distinction between patients and healthy controls for each disorder, we categorized participants into one of five groups in a single analysis: healthy control, Dysarthria, Dysglossia, Dysphonia, and CLP. To maintain fairness, we included an equal number of speakers from each category, based on the smallest subgroup, Dysphonia, which comprised $n = 78$ speakers. Consequently, we selected 78 speakers from each category, dividing them into a training set of $n = 372$ speakers (62 from each disorder, 62 from adult controls, and 62 from child controls) and a test set of $n = 96$ speakers (16 from each). This approach ensured each category was equally represented, with test speakers excluded from the training phase to guarantee they were unseen during the evaluation.

**Exploring inversion methods in speech anonymization.** In the final step of our investigation, we examined potential inversion risks within ASV systems. While membership inference attacks[62] and counter-measures like differential privacy[63,64] are well-discussed in the image processing domain[59,61], their implications for speech data anonymization are less explored[65,66]. Specifically, we investigated how well the randomized McAdams coefficient method could stand against inversion attempts that aim to reverse the anonymization and identify speakers.

Considering a scenario where an external party is aware of the anonymization system's specifics, including the McAdams coefficients, they might attempt to exploit this knowledge. This could involve training a counter ASV system tailored to recognize speakers despite their speech being anonymized. To test this, we utilized the same subset of the LibriSpeech[37] dataset previously employed for our primary ASV system training, aiming for a straightforward comparison. This phase included training with both original and anonymized speech samples, using the randomized McAdams coefficient method, where anonymized versions were considered authentic utterances of the speakers. This setup helped us assess the feasibility of linking anonymized voices back to their original speakers, providing initial insights into our anonymization technique's resistance to inversion efforts.

**Generalization of the method beyond German language.** The anonymization methods presented are not reliant on language-specific characteristics, demonstrating their adaptability across languages. We validated this generalization using the PC-GITA dataset[67], which consists of speech recordings from $n = 50$ Parkinson's Disease (PD) patients and $n = 50$ matched healthy controls (by age and gender), all native Spanish speakers from Colombia. The recordings were collected in accordance with a protocol designed to meet technical specifications and recommendations from experts in linguistics, phoniatry, and neurology. All recordings were made in controlled noise conditions within a soundproof booth, and the audio was sampled at 44.1 kHz with 16-bit resolution. None of the healthy controls exhibited symptoms of PD or any other neurological disorders.

The protocol for the PC-GITA dataset[67] was approved by the Ethical Committee of the Research Institute in the Faculty of Medicine at the University of Antioquia in Medellín, Colombia (approval 19-63-673). All experiments were conducted in accordance with applicable national and international guidelines and regulations. Informed consent was obtained from all adult participants, as well as from the parents or legal guardians of the children involved. Our use of the PC-GITA dataset did not require separate ethical approval, as it is a restricted-access resource. Access was granted following our agreement to the dataset's user terms.

For anonymization, we employed the McAdams Coefficient method, similar to that used with the German dataset. We utilized phonemic place of articulation features to distinguish between PD patients and healthy controls. A linear support vector regression machine[68] was applied to predict the maximum phonation duration. The utility of the method was quantitatively assessed using Pearson's r correlation coefficient, comparing the PD patients and healthy controls.

## Hardware
The hardware used in our experiments included Intel CPUs with 18 and 32 cores, 32 GB of RAM, and Nvidia GPUs such as the GeForce GTX 1080 Ti (11 GB), V100 (16 GB), RTX 6000 (24 GB), Quadro 5000 (32 GB), and Quadro 6000 (32 GB).

## Reporting summary
Further information on research design is available in the Nature Portfolio Reporting Summary linked to this article.

## Results
### Impact of anonymization on pathological speech biomarkers
Table 2 presents the effects of anonymization on pathological speech biomarkers, illustrating a notable rise in EER following anonymization,

signaling improved privacy measures. Figure 2 displays frequency spectrums and power spectral densities (PSD) of both original and anonymized speech signals, offering a sample utterance from each disorder for illustrative purposes. We note that anonymization leads to a reduction in the signal's power, indicating that anonymized signals exhibit lower PSD compared to their original counterparts. This observation suggests that anonymization not only obscures speaker identity but may also affect the acoustic properties of speech.

The anonymization performance varied by disorder type, with the randomized McAdams coefficient anonymization method outperforming the DL-based (i.e., the randomized pitch shift algorithm) anonymization method. Initially, EER values for disorders such as Dysarthria, Dysglossia, Dysphonia, and CLP were $1.80 \pm 0.42\%$, $1.78 \pm 0.43\%$, $2.19 \pm 0.30\%$, and $7.01 \pm 0.24\%$, respectively. After employing the randomized pitch shift algorithm, these values escalated to $30.72 \pm 0.48\%$, $31.54 \pm 0.87\%$, $41.02 \pm 0.33\%$, and $38.73 \pm 0.39\%$, showcasing increases of 1606%, 1672%, 1773%, and 452%, respectively. Similarly, the randomized McAdams coefficient method elevated EERs to $36.59 \pm 0.64\%$, $34.26 \pm 0.67\%$, $38.86 \pm 0.35\%$, and $32.19 \pm 0.46\%$, indicating equivalent percentage increases.

Regarding utility, or the networks' ability to detect disorders, the DL-based method notably compromised utility across all disorders. Conversely, the McAdams coefficient method resulted in a decrease in AUROC for Dysarthria ($p = 5.5 \times 10^{-27}$), with only a modest decrease of 2.60% in AUROC post-anonymization. Dysphonia and CLP experienced AUROC reductions of 0.75% ($p = 3.4 \times 10^{-13}$) and 0.07% ($p = 0.14$), respectively. Notably, for Dysglossia, anonymization via the McAdams coefficient method led to a significant increase ($p = 6.1 \times 10^{-21}$) of 1.11% in AUROC.

Following these insights, we refined our examination of the DL-based method by omitting the pitch shifting process to scrutinize its synthesization phase's impact. This phase involved reconstructing an input utterance solely with the vocoder module. The findings, detailed in Table 2, reveal a nuanced effect: while Dysglossia showed a slight improvement in AUROC ($97.86 \pm 0.33\%$ vs. $97.73 \pm 0.41\%$, $p = 0.097$), significant reductions were observed for other disorders without enhancing anonymization substantially. This indicates that utility is influenced not just by the identity-altering pitch shifting process but also by the synthesization phase itself.

### Privacy-utility tradeoff results
Given the McAdams Coefficient method's effectiveness over the DL-based approach, we further investigated this method exclusively for the remainder of our experiments. We adjusted the McAdams coefficient method to utilize fixed values, embarking on a series of controlled experiments. As depicted in Fig. 3, we observed that a linear increase in the coefficient value led to a logarithmic decrease in EER across all studied disorders, indicating that while the method supports customizable levels of anonymization, coefficient adjustments impact the effectiveness of anonymization in a logarithmic manner.

In terms of utility, adjusting anonymization levels (reflected by varying EER) had diverse effects on the network's diagnostic capabilities across different disorders. Supplementary Table 2 elaborates on the AUROC, accuracy, sensitivity, and specificity metrics at different coefficient settings. Consistent with our early observations, Dysphonia showed a marked reduction in AUROC following anonymization ($p = 1.0 \times 10^{-31}$). Surprisingly, adjustments in anonymization levels did not produce a uniform trend in AUROC across all disorders, suggesting that the anonymization level's impact on diagnostic accuracy is disorder-specific. The Pearson correlation coefficients between EER and AUROC values ($-0.613$ for Dysarthria, $-0.112$ for Dysglossia, 0.425 for Dysphonia, and $-0.397$ for CLP; with corresponding $p$-values of 0.19, 0.83, 0.40, and 0.44) were not statistically significant. Optimal privacy-utility balance was achieved with coefficients ranging from 0.75 to 0.9, corroborated by both EER and classification metrics. Subjective analysis of the anonymized speech waveforms affirmed that coefficients lower than 0.75 compromised audio quality, impacting not only pathological markers but also overall speech clarity.

**Table 2 | Evaluation results of randomized anonymization methods on pathological speech**

| | | Dysarthria | Dysglossia | Dysphonia | Cleft Lip and Palate |
|---|---|---|---|---|---|
| EER [%] | Original | $1.80 \pm 0.42$ | $1.78 \pm 0.43$ | $2.19 \pm 0.30$ | $7.01 \pm 0.24$ |
| | Pitch Shift + HiFi-GAN | $30.72 \pm 0.48$ | $31.54 \pm 0.87$ | $41.02 \pm 0.33$ | $38.73 \pm 0.39$ |
| | Only HiFi-GAN | $7.48 \pm 0.55$ | $11.39 \pm 1.17$ | $11.27 \pm 0.40$ | $23.45 \pm 0.27$ |
| | McAdams | $36.59 \pm 0.64$ | $34.26 \pm 0.67$ | $38.86 \pm 0.35$ | $32.19 \pm 0.46$ |
| AUROC [%] | Original | $97.33 \pm 0.51$ | $97.73 \pm 0.41$ | $99.12 \pm 0.42$ | $96.44 \pm 0.21$ |
| | Pitch Shift + HiFi-GAN | $90.56 \pm 0.66$ ($p = 1.0 \times 10^{-45}$) | $95.89 \pm 0.49$ ($p = 4.6 \times 10^{-25}$) | $97.14 \pm 0.33$ ($p = 5.8 \times 10^{-30}$) | $93.20 \pm 0.23$ ($p = 3.7 \times 10^{-50}$) |
| | Only HiFi-GAN | $96.50 \pm 0.61$ ($p = 1.9 \times 10^{-9}$) | $97.86 \pm 0.33$ ($p = 0.097$) | $98.50 \pm 0.30$ ($p = 5.8 \times 10^{-11}$) | $90.50 \pm 0.44$ ($p = 3.1 \times 10^{-56}$) |
| | McAdams | $94.86 \pm 0.59$ ($p = 5.5 \times 10^{-27}$) | $98.86 \pm 0.28$ ($p = 6.1 \times 10^{-21}$) | $98.38 \pm 0.31$ ($p = 3.4 \times 10^{-13}$) | $96.37 \pm 0.28$ ($p = 0.14$) |
| Accuracy [%] | Original | $93.80 \pm 0.73$ | $92.87 \pm 0.84$ | $97.37 \pm 0.59$ | $90.99 \pm 0.44$ |
| | Pitch Shift + HiFi-GAN | $83.51 \pm 1.15$ ($p = 2.8 \times 10^{-44}$) | $89.90 \pm 1.20$ ($p = 8.6 \times 10^{-19}$) | $91.18 \pm 0.70$ ($p = 5.1 \times 10^{-42}$) | $84.66 \pm 0.36$ ($p = 5.6 \times 10^{-51}$) |
| | Only HiFi-GAN | $91.97 \pm 0.99$ ($p = 6.6 \times 10^{-14}$) | $93.39 \pm 0.80$ ($p = 0.0020$) | $93.96 \pm 0.73$ ($p = 1.5 \times 10^{-29}$) | $85.45 \pm 0.64$ ($p = 5.2 \times 10^{-47}$) |
| | McAdams | $90.67 \pm 0.94$ ($p = 2.2 \times 10^{-23}$) | $95.56 \pm 0.60$ ($p = 3.1 \times 10^{-23}$) | $94.73 \pm 0.67$ ($p = 1.3 \times 10^{-25}$) | $91.14 \pm 0.51$ ($p = 0.12$) |
| Sensitivity [%] | Original | $94.11 \pm 1.05$ | $92.55 \pm 1.51$ | $97.10 \pm 0.71$ | $90.22 \pm 0.98$ |
| | Pitch Shift + HiFi-GAN | $86.64 \pm 2.21$ ($p = 3.5 \times 10^{-26}$) | $89.66 \pm 1.87$ ($p = 4.7 \times 10^{-11}$) | $91.61 \pm 1.74$ ($p = 2.6 \times 10^{-25}$) | $86.64 \pm 2.21$ ($p = 2.3 \times 10^{-22}$) |
| | Only HiFi-GAN | $90.60 \pm 1.57$ ($p = 2.2 \times 10^{-17}$) | $92.82 \pm 1.50$ ($p = 0.93$) | $94.15 \pm 1.27$ ($p = 8.4 \times 10^{-19}$) | $84.78 \pm 1.39$ ($p = 2.0 \times 10^{-31}$) |
| | McAdams | $89.20 \pm 1.63$ ($p = 9.4 \times 10^{-23}$) | $95.13 \pm 0.97$ ($p = 2.1 \times 10^{-13}$) | $94.50 \pm 1.02$ ($p = 2.6 \times 10^{-19}$) | $90.76 \pm 1.01$ ($p = 0.0092$) |
| Specificity [%] | Original | $92.68 \pm 1.03$ | $93.49 \pm 1.36$ | $97.70 \pm 0.76$ | $91.64 \pm 1.07$ |
| | Pitch Shift + HiFi-GAN | $79.52 \pm 2.72$ ($p = 8.0 \times 10^{-34}$) | $89.55 \pm 1.75$ ($p = 1.2 \times 10^{-16}$) | $90.91 \pm 1.48$ ($p = 1.0 \times 10^{-31}$) | $79.52 \pm 2.72$ ($p = 3.0 \times 10^{-31}$) |
| | Only HiFi-GAN | $92.79 \pm 1.56$ ($p = 0.68$) | $93.33 \pm 1.14$ ($p = 0.20$) | $93.85 \pm 1.78$ ($p = 1.7 \times 10^{-18}$) | $84.53 \pm 1.33$ ($p = 5.3 \times 10^{-37}$) |
| | McAdams | $90.36 \pm 1.43$ ($p = 3.7 \times 10^{-12}$) | $95.28 \pm 1.03$ ($p = 2.1 \times 10^{-9}$) | $95.02 \pm 1.32$ ($p = 1.8 \times 10^{-16}$) | $92.42 \pm 0.91$ ($p = 0.00028$) |

This table presents the outcomes of applying randomized anonymization methods to pathological speech, with the anonymization level gauged by equal error rate (EER) values. The results are presented as mean ± standard deviation. The utility for detecting disorders such as Dysarthria, Dysglossia, Dysphonia, and Cleft Lip and Palate is evaluated, using metrics including the area under the receiver operating characteristic curve (AUROC), accuracy, sensitivity, and specificity. Statistical significance between original and anonymized data for these utility metrics was determined using a two-tailed unpaired t-test, with p-values noted. Original denotes data prior to anonymization. Pitch Shift + HiFi-GAN represents the randomized pitch shift method based on Supplementary Fig. 1, which is a deep-learning-based anonymization method. Only HiFi-GAN indicates a simplified version of the randomized pitch shift method where no modifications are applied to log-Mel-spectrograms before synthesizing them using the HiFi-GAN vocoder[57]. McAdams details the randomized McAdams coefficient method for anonymization. Training sets comprised $n = 168$ speakers (Dysarthria detection), $n = 168$ (Dysglossia detection), $n = 110$ (Dysphonia detection), and $n = 887$ (CLP detection). Corresponding test sets included $n = 73$ (Dysarthria detection), $n = 73$ (Dysglossia detection), $n = 49$ (Dysphonia detection), and $n = 381$ (CLP detection).

## Privacy-fairness tradeoff results

Table 3 presents our analysis on how anonymization affects the balance between privacy and fairness across different demographics. In terms of privacy, we observed a uniform increase in EER across gender subgroups, except in the case of Dysphonia detection. Here, female speakers initially had a significantly lower EER compared to males ($0.22 \pm 0.22\%$ vs $2.28 \pm 0.32\%$), but anonymization leveled the playing field, bringing both genders to similar privacy levels ($35.09 \pm 1.31\%$ for females and $37.99 \pm 0.35\%$ for males). Regarding age subgroups, adults and children started with EERs of $1.25 \pm 0.29$ and $6.17 \pm 0.24\%$, respectively. Post-anonymization, both groups converged to approximately 32% EER ($32.26 \pm 0.31\%$ for adults and $32.08 \pm 0.50\%$ for children), indicating that anonymization effectively equalizes privacy across ages as well.

Regarding fairness in performance (Fig. 4), minor disparities were noted. Dysarthria detection showed a slight AUROC reduction for females (around 1%) compared to males (around 4%). This discrepancy is reflected in the statistical parity difference (PtD), which increased from 0.02 to 0.04

for females, post-anonymization. A similar pattern emerged in CLP detection.

For Dysglossia and Dysphonia, disparities were more pronounced, with PtD changes around 0.04 but in opposite directions. In Dysglossia, the network initially favored females (PtD = 0.03), shifting to slightly favor males (PtD = 0.01) post-anonymization. Conversely, in Dysphonia, the initial advantage for males (PtD = 0.02) switched to favor females (PtD = 0.02) after anonymization.

Age-related analysis showed nearly consistent performance post-anonymization. Initially favoring children in general disorder detection (PtD = 0.03), the advantage slightly decreased to PtD=0.02 but remained in favor of children.

## Efficacy of automatic anonymization in large and diverse pathological datasets

Upon aggregating all patient and control data into comprehensive groups ($n = 1333$ for training and $n = 576$ for test), we noted a substantial increase

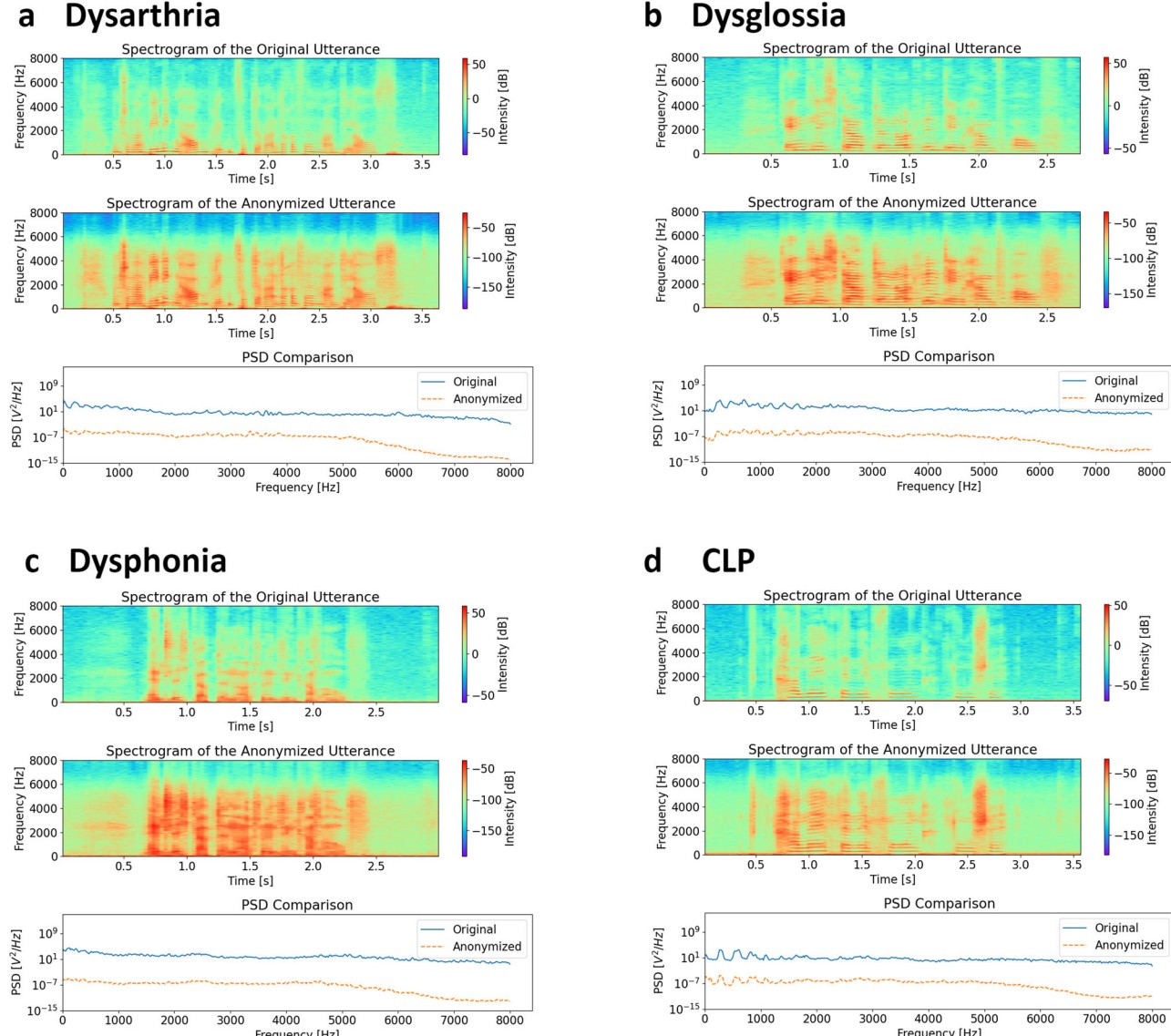

**Fig. 2 | Spectral representation of original vs. anonymized speech signals.** The frequency spectrums and power spectral densities (PSD) of both original and anonymized speech signals. An exemplary sample utterance from each disorder including (**a**) Dysarthria, (**b**) Dysglossia, (**c**) Dysphonia, and (**d**) Cleft Lip and Palate (CLP) is shown.

in EER post-anonymization: from $2.96 \pm 0.10\%$ to $30.24 \pm 0.33\%$ for patients, from $5.20 \pm 0.11\%$ to $31.61 \pm 0.13\%$ for healthy controls, and from $4.02 \pm 0.02\%$ to $32.77 \pm 0.05\%$ for all data. This indicates comparable anonymization effectiveness across both patient and control groups, with EERs rising approximately 27% despite the initial patient EER being roughly 2 times lower than that of controls.

Regarding utility, the change was statistically significant ($p = 1.2 \times 10^{-61}$), with AUROC showing a minimal decrease of less than 1% from $97.05 \pm 0.16\%$ to $96.07 \pm 0.19\%$ post-anonymization, suggesting a negligible impact on the ability to detect disorders. Figure 5 depicts these effects on utility, showcasing metrics such as AUROC, accuracy, sensitivity, and specificity, indicating that anonymization can substantially enhance privacy without substantially compromising diagnostic utility.

**Multiclass classification performance**
Figure 6 illustrates the outcomes of our multiclass classification experiment, posing a more complex challenge than prior binary classifications. In this setup, the model distinguishes whether an utterance is healthy or indicative of Dysarthria, Dysglossia, Dysphonia, or CLP. Drawing from the binary classification insights (see Fig. 3), we narrowed the McAdams coefficient to the [0.7, 1.0] range, where its effectiveness peaked, while halving the

increment steps for greater precision. Overall, results echoed binary classification trends, with a notable exception: Dysphonia, previously showing lower AUROC scores for anonymized data, now demonstrated improved AUROC values.

A critical observation from this experiment is that while a strict monotonic relationship between privacy levels and utility remains elusive, specifying ranges for each disorder reveals potential for monotonicity. This insight underscores that the privacy-utility tradeoff is substantially influenced by the specific disorder in question, with the optimal balance being unique to each condition.

**Inversion attack outcomes**
In assessing the inversion attack's outcomes, our experiment revealed EER values for disorders such as Dysarthria, Dysglossia, Dysphonia, and CLP at $1.64 \pm 0.24\%$, $1.58 \pm 0.29\%$, $1.63 \pm 0.12\%$, and $5.86 \pm 0.20\%$, respectively, using the inverse ASV system. After applying the randomized McAdams coefficient method for anonymization, these EERs rose to $7.08 \pm 0.40\%$, $8.66 \pm 0.63\%$, $10.48 \pm 0.42\%$, and $12.00 \pm 0.31\%$, respectively. Despite the substantial increase in EER post-anonymization, indicating a heightened level of privacy, the inverse ASV system substantially compromised the anonymization's efficacy. For the

**Fig. 3 | Analysis of privacy-utility tradeoff.** The X-axis represents varying levels of the McAdams coefficient. The utility of the pathological speech data for disorder classification is quantified using the area under the receiver operating characteristic curve (AUROC) values for each disorder. The effectiveness of anonymization (privacy) is evaluated through equal error rate (EER) values. Whiskers show the error bars, which indicate the standard deviation values. Notably, an increase in the McAdams coefficient corresponds to a logarithmic decline in EER, indicating enhanced privacy. However, the AUROC values do not follow a consistent trend across the coefficient range. The findings from the (**a**) Dysarthria detection task, (**b**) Dysglossia detection task, (**c**) Dysphonia detection task, and (**d**) Clef Lip and Palate (CLP) detection task are illustrated. Training sets comprised n = 168 speakers (Dysarthria detection), n = 168 (Dysglossia detection), n = 110 (Dysphonia detection), and n = 887 (CLP detection). Corresponding test sets included n = 73 (Dysarthria detection), n = 73 (Dysglossia detection), n = 49 (Dysphonia detection), and n = 381 (CLP detection). Source data is in Supplementary Data 1.

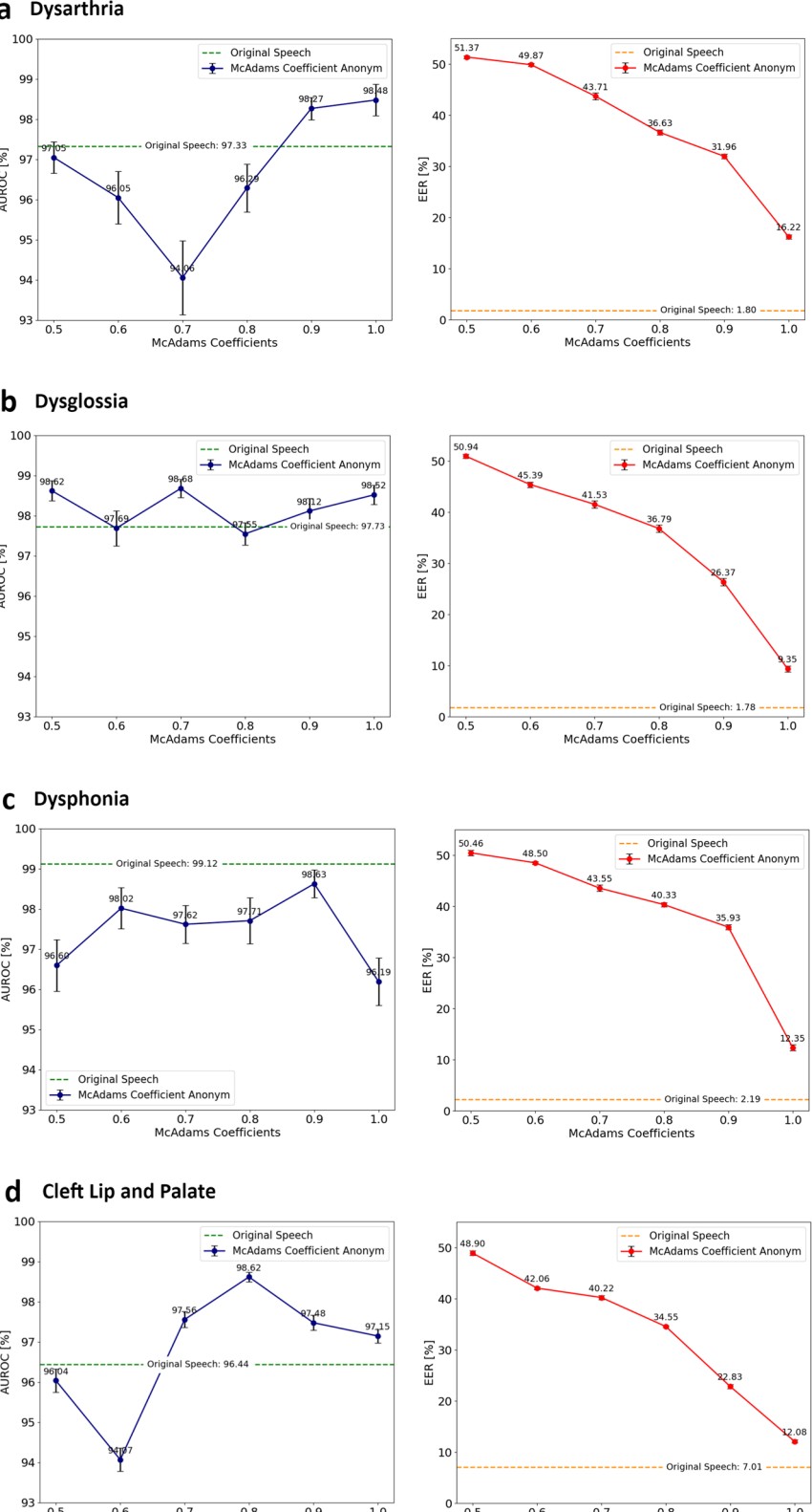

primary ASV method compared to the inverse ASV system, the percentage increase in EER post-anonymization was noted as 1933% vs. 332% for Dysarthria, 1825% vs. 448% for Dysglossia, 1674% vs. 543% for Dysphonia, and 359% vs. 105% for CLP. This pattern was evident in both pathological and healthy speech, with EERs for healthy adults rising from 2.66 ± 0.39% to 8.26 ± 0.50%, and for children from 7.37 ± 0.16% to

14.90 ± 0.20% after anonymization, using the inverse ASV system. These results indicate that speech anonymization methods, whether applied to healthy or pathological speech, may not fully withstand inversion attacks. This underscores the necessity for ongoing research into strengthening the resilience of these methods, ensuring comprehensive privacy protection in speech data applications.

**Table 3 | Privacy-fairness tradeoff in pathological speech anonymization**

| Disorder | Metric | FEMALE | | MALE | |
|---|---|---|---|---|---|
| | | Original | Anonymized | Original | Anonymized |
| Dysarthria | EER [%] | 2.22 ± 0.29 | 40.77 ± 0.51 | 2.17 ± 0.87 | 36.65 ± 0.84 |
| | AUROC [%] | 98.98 ± 0.30 | 97.83 ± 0.62 ($p = 1.3 \times 10^{-15}$) | 96.82 ± 0.86 | 92.78 ± 1.04 ($p = 8.0 \times 10^{-26}$) |
| | PtD | 0.02 | 0.04 | −0.02 | −0.04 |
| Dysglossia | EER [%] | 1.71 ± 0.93 | 35.21 ± 1.59 | 1.95 ± 0.47 | 37.14 ± 0.78 |
| | AUROC [%] | 98.74 ± 0.64 | 98.93 ± 0.36 ($p = 0.083$) | 97.50 ± 0.54 | 98.59 ± 0.44 ($p = 1.1 \times 10^{-14}$) |
| | PtD | 0.03 | −0.01 | −0.03 | 0.01 |
| Dysphonia | EER [%] | 0.22 ± 0.22 | 35.09 ± 1.31 | 2.28 ± 0.32 | 37.99 ± 0.35 |
| | AUROC [%] | 98.48 ± 0.78 | 99.32 ± 0.36 ($p = 1.2 \times 10^{-8}$) | 99.77 ± 0.14 | 97.49 ± 0.59 ($p = 5.1 \times 10^{-30}$) |
| | PtD | −0.02 | 0.02 | 0.02 | −0.02 |
| CLP | EER [%] | 7.97 ± 0.36 | 32.75 ± 0.57 | 5.39 ± 0.22 | 32.07 ± 0.59 |
| | AUROC [%] | 96.75 ± 0.27 | 97.26 ± 0.30 ($p = 1.1 \times 10^{-11}$) | 96.23 ± 0.26 | 95.28 ± 0.38 ($p = 3.4 \times 10^{-19}$) |
| | PtD | 0.01 | 0.03 | −0.01 | −0.03 |
| Disorder | Metric | ADULTS | | CHILDREN | |
| | | Original | Anonymized | Original | Anonymized |
| General Speech & Voice Disorder | EER [%] | 1.25 ± 0.29 | 32.26 ± 0.31 | 6.17 ± 0.24 | 32.08 ± 0.50 |
| | AUROC [%] | 94.72 ± 0.49 | 94.14 ± 0.52 ($p = 9.3 \times 10^{-7}$) | 97.47 ± 0.20 | 96.67 ± 0.24 ($p = 6.9 \times 10^{-23}$) |
| | PtD | −0.03 | −0.02 | 0.03 | 0.02 |

This table compares the original and anonymized pathological speech data, focusing on the anonymization level and diagnostic fairness across different demographic subgroups. It evaluates the utility of disorder classification following the randomized McAdams coefficient anonymization method, measured by the area under the receiver operating characteristic curve (AUROC), and assesses privacy through equal error rate (EER) within these subgroups. The AUROC and EER values are presented as mean ± standard deviation. The analysis of demographic fairness is conducted using the statistical parity difference (PtD), where positive values indicate a benefit to the minority class, and negative values suggest discrimination against it. Examined demographic subgroups include gender (female and male) for disorders such as Dysarthria, Dysglossia, Dysphonia, and Cleft Lip and Palate (CLP), and age (adults and children) for broad speech and voice disorder detection. A two-tailed unpaired t-test was used to evaluate statistical significance between original and anonymized datasets in terms of AUROC, with p-values provided. Training sets comprised $n = 168$ speakers (Dysarthria detection), $n = 168$ (Dysglossia detection), $n = 110$ (Dysphonia detection), and $n = 887$ (CLP detection). Corresponding full test sets included $n = 73$ (Dysarthria detection), $n = 73$ (Dysglossia detection), $n = 49$ (Dysphonia detection), and $n = 381$ (CLP detection). Female test sets included $n = 37$ (Dysarthria detection), $n = 19$ (Dysglossia detection), $n = 18$ (Dysphonia detection), and $n = 187$ (CLP detection). Male test sets included $n = 36$ (Dysarthria detection), $n = 54$ (Dysglossia detection), $n = 31$ (Dysphonia detection), and $n = 194$ (CLP detection). The adults test set included $n = 171$ (General Speech & Voice Disorder detection) and the children test set included $n = 381$ (General Speech & Voice Disorder detection).

## Generalization to other languages

Supplementary Fig. 2 presents the results of both utility and privacy assessments using the PC-GITA dataset[67], which includes Spanish-speaking PD patients. The overall EER of the anonymized PD dataset is 34.00%, comparable to the results from the German dataset, where the EERs for Dysarthria, Dysglossia, Dysphonia, and CLP are 36.59%, 34.26%, 38.86%, and 32.19%, respectively. Similarly, a logarithmic decline in EER values is observed with linear increases in the McAdams coefficient, akin to the German dataset. Notably, the magnitude of EER changes increases with further adjustments to the coefficient, underscoring the necessity for disorder-specific configurations in anonymization methods. Like in the German dataset, the anonymization process uniformly conceals the identities of both patients and healthy controls. Initially, the EER for controls is 2.00%, while for PD it is 3.78% (nearly twice as high). After anonymization using a random coefficient, the EER for controls and PD equalizes at 34.9% and 34%, respectively, demonstrating that the initial disparity in identification difficulty is effectively neutralized.

Regarding utility, the original data exhibited a correlation coefficient of 0.71. Post-anonymization, this value decreases to 0.42 with a randomized coefficient and to 0.57 with a fixed coefficient set at 0.8. These findings indicate a reduction in some pathological biomarkers, yet the results align well with those from the German dataset. This alignment supports the generalizability of the proposed methods, while also highlighting the critical need for continued research into disorder-specific anonymization techniques.

## Discussion

Our investigation into the impact of anonymization on pathological speech biomarkers across a dataset of over 2700 speakers revealed substantial privacy enhancements, especially utilizing the McAdams Coefficient method and DL-based approaches, as evidenced by increased EER. Despite this improvement in privacy, the effect on the utility for diagnosing specific speech and voice disorders varied, maintaining overall minimal influence on diagnostic accuracy. Notably, anonymization had a modest impact on Dysarthria, Dysphonia, and CLP, yet interestingly, it benefited Dysglossia. This advantage for Dysglossia could stem from its primary manifestation in articulatory changes affecting vowels and consonants, suggesting a lesser susceptibility to the anonymization process's alterations.

Our research underscores that automatic anonymization consistently affects the utility of pathological speech data across different methodologies. Despite this, the tradeoff observed between the level of anonymization and the utility of pathological speech data leans towards a positive equilibrium, underscoring the effectiveness of modern anonymization techniques in managing pathological speech. This balance is particularly pronounced in conditions such as Dysarthria, Dysglossia, Dysphonia, and CLP, highlighting the practicality of these anonymization strategies in preserving the integrity of medical diagnostics while enhancing privacy.

We evaluated both DL-based and signal-level modification anonymization methods for their applicability to pathological speech. The signal-level modification methods, particularly the McAdams Coefficient method, emerged as generally superior to DL-based approaches in anonymizing

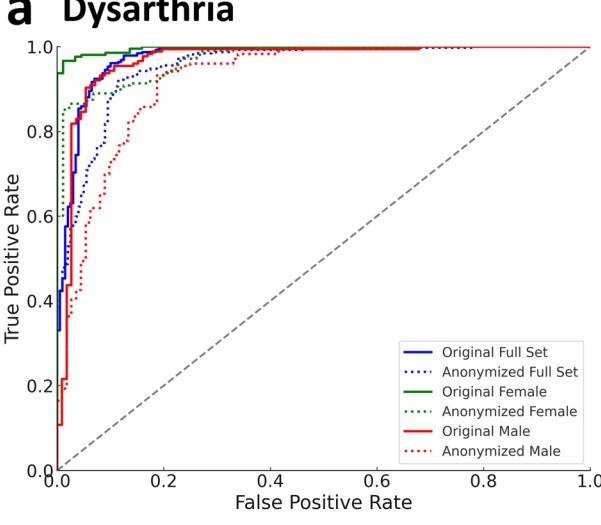

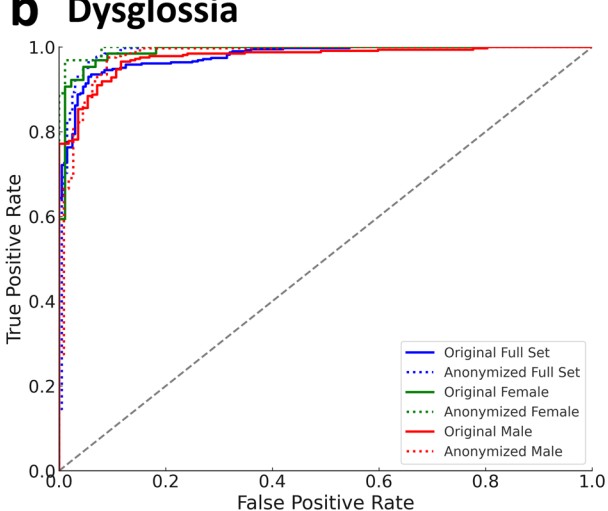

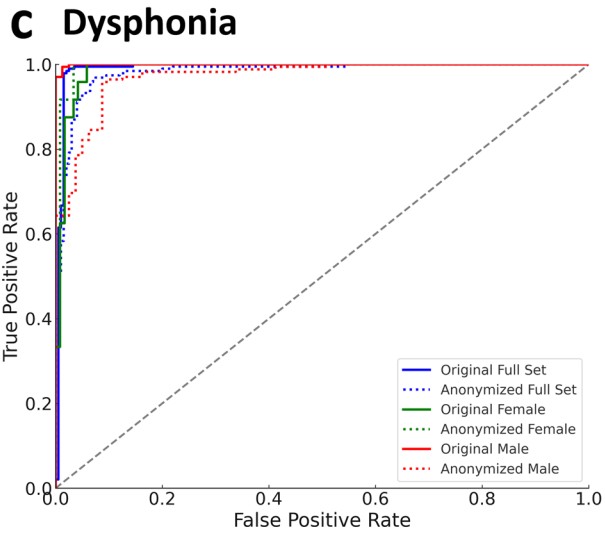

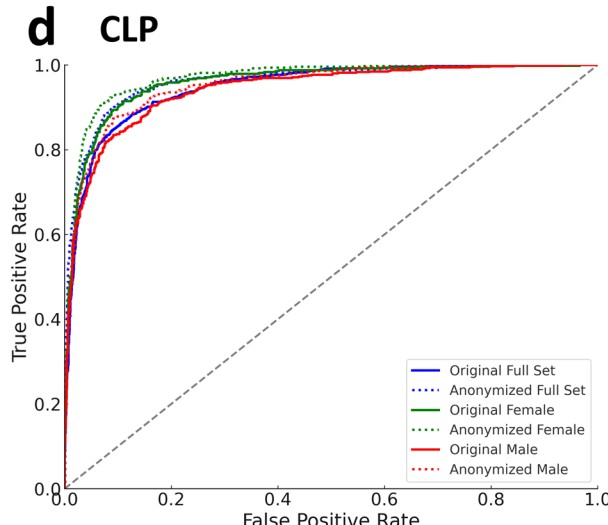

**Fig. 4 | Receiver operating characteristic (ROC) curves for original vs. anonymized speech data across gender subgroups.** This figure displays the ROC curves comparing the performance of models on original and anonymized speech data, employing the randomized McAdams coefficient anonymization method, for (**a**) Dysarthria, (**b**) Dysglossia, (**c**) Dysphonia, and (**d**) Cleft Lip and Palate (CLP). Solid lines represent models using original data, while dotted curves indicate models using anonymized data. Performance on the entire dataset is depicted in blue, with green and red highlighting the results for male and female subgroups, respectively. The axes plot the True Positive Rate (sensitivity) against the False Positive Rate (1-specificity), with a diagonal gray line marking the threshold of no discrimination.

Training sets comprised $n = 168$ speakers (Dysarthria detection), $n = 168$ (Dysglossia detection), $n = 110$ (Dysphonia detection), and $n = 887$ (CLP detection). Corresponding full test sets included $n = 73$ (Dysarthria detection), $n = 73$ (Dysglossia detection), $n = 49$ (Dysphonia detection), and $n = 381$ (CLP detection). Female test sets included $n = 37$ (Dysarthria detection), $n = 19$ (Dysglossia detection), $n = 18$ (Dysphonia detection), and $n = 187$ (CLP detection). Male test sets included $n = 36$ (Dysarthria detection), $n = 54$ (Dysglossia detection), $n = 31$ (Dysphonia detection), and $n = 194$ (CLP detection). Source data is in Supplementary Data 1.

pathological speech. The DL-based method we utilized, which aligns with the broader category of voice conversion methods, adopted a simplified strategy. Instead of converting speaker identity, we applied a randomized pitch shift followed by speech synthesization. This approach was specifically chosen to facilitate the anonymization of a vast array of 2742 unique speakers, aiming to enhance the availability of public datasets for the development of data-driven pathological speech analysis tools. This focus on a real-world application underscores the limited utility of voice conversion methods that merely alter a speaker's identity to another in our context. Future investigations might delve into the nuances of more advanced voice conversion-based anonymization methods for pathological speech, examining their specific advantages and applications, while mindful

that alterations in fundamental frequency (F0) and formants could obscure critical pathological speech characteristics necessary for in-depth analysis.

Our analysis of the privacy-utility tradeoff has yielded substantial insights. Contrary to common assumptions[25], we found that the relationship between the level of anonymization and its utility is not strictly monotonic. Adjusting the McAdams coefficient linearly leads to logarithmic variations in EER, affecting the identification capabilities for various disorders in distinct ways. Importantly, an increase in EER does not universally diminish the utility of pathological speech data; rather, the impact varies by disorder. We observed that for each type of speech or voice disorder, there exists a specific anonymization level that offers a more favorable privacy-utility balance. This underscores the importance of pinpointing the optimal

**Fig. 5 | Utility comparison of original versus anonymized speech data on the entire dataset.** We consolidated all patient data into a general patient set and all control data into a general control set other ($n = 1333$ training speakers and $n = 576$ test speakers) and performed a general speech & voice disorder detection. The utility of the outcomes of applying the randomized McAdams coefficient anonymization method to the pathological speech data for disorder classification is quantified using (**a**) the area under the receiver operating characteristic curve (AUROC), (**b**) accuracy, (**c**) sensitivity, and (**d**) specificity values. Box plots display the distribution of each evaluation metric along the y-axis, with the corresponding ranges shown on the x-axis. Crosses on the plot represent the means, while the boxes depict the interquartile range from the first quartile (Q1) to the third quartile (Q3), with a central line marking the median. Whiskers stretch to 1.5 times the interquartile range beyond Q3 and below Q1. Data points falling outside this span are labeled as outliers (represented by dots). Source data is in Supplementary Data 1.

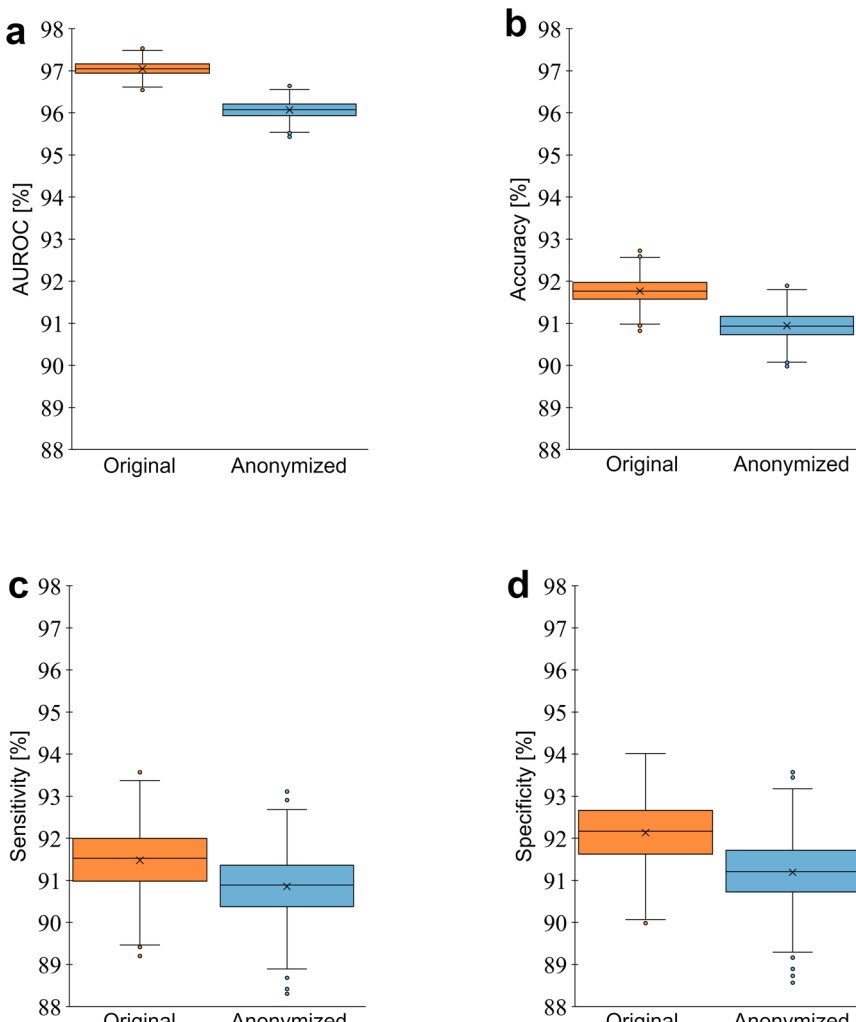

tradeoff point for each disorder to achieve a balanced enhancement of privacy while maintaining utility.

Despite limited exploration of anonymization in pathological speech, research on privacy-preserving pathological AI models within imaging has highlighted potential effects on demographic fairness[61,69,70]. Our study extended this investigation to the speech domain, evaluating the privacy-fairness tradeoff. We observed uniform anonymization efficacy across demographic subgroups, with minor diagnostic fairness discrepancies, especially notable in gender differences within Dysphonia detection. Initially, females had substantially lower EER than males, yet post-anonymization, both genders achieved similar privacy levels. A deeper look revealed an underrepresentation of females ($n = 8$) compared to males ($n = 70$) in this subgroup. The overall impact on diagnostic fairness was modest, typically under 4%, and varied by disorder. In Dysarthria and CLP, minimal effects occurred, whereas in Dysglossia and Dysphonia a shift in which gender subgroup was favored post-anonymization can be seen. Age subgroup analysis showed nearly uniform impacts, underscoring the nuanced influence of anonymization on demographic fairness.

Leveraging a diverse dataset, we assessed the anonymization's effectiveness across a combined set of patient and control data. This approach aimed to challenge the anonymization methods with a broad spectrum of speech characteristics, in line with findings[13] that diversity in speaker and disorder profiles can complicate anonymization. Our results affirm the feasibility of applying automatic anonymization to extensive pathological speech datasets, enhancing privacy with minimal impact on the clinical utility of speech data for diagnostics.

Determining a safe level of anonymity involves assessing the risk of identification within a dataset. Moreover, we acknowledge that absolute privacy, defined as zero risk, can only be achieved when no information is disclosed[71]. For the entire dataset of $n = 2742$ speakers, the original EER stood at $4.02 \pm 0.02\%$. Assuming all speakers are published, and an individual from this dataset attempts to re-identify their recording, the breakdown is as follows: false acceptance (FA) = $2741 \times 4.02\% = 110.19 \approx 110$ (rounded), true acceptance (TA) = $1 \times 95.98\% = 0.960$, false rejection (FR) = $1 \times 4.02\% = 0.040$, and true rejection (TR) = $2741 \times 95.98\% = 2630.810$. This scenario yields approximately 110 recordings requiring manual verification, translating to a 1:110 chance of accurate identification. Post-anonymization, with an EER increase to $32.77 \pm 0.05\%$, the likelihood adjusts to 1:898 ($2741 \times 30.24\% = 898.230$), marking a substantial improvement in privacy. However, the practicality of manually sifting through such a large number of potential matches is constrained by computational limitations. Thus, this level of anonymization is deemed sufficient.

This effect even becomes more pronounced when we focus solely on publishing the patient subset ($n = 1443$) instead of the entire dataset. Initially, with an EER of $2.96 \pm 0.10\%$, the chance of identification before anonymization stood at 1:43, a range feasibly manageable for manual checking. Post-anonymization, with the EER escalating to $30.24 \pm 0.33\%$, this chance dramatically improves to 1:436, exemplifying an ideal enhancement in privacy. When focusing on a smaller dataset, such as the Dysphonia subset with $n = 78$ subjects, the original EER of $2.19 \pm 0.30\%$ increased to $38.86 \pm 0.35\%$ after anonymization. This change boosts the identification challenge from 1:2 in the original dataset to 1:30

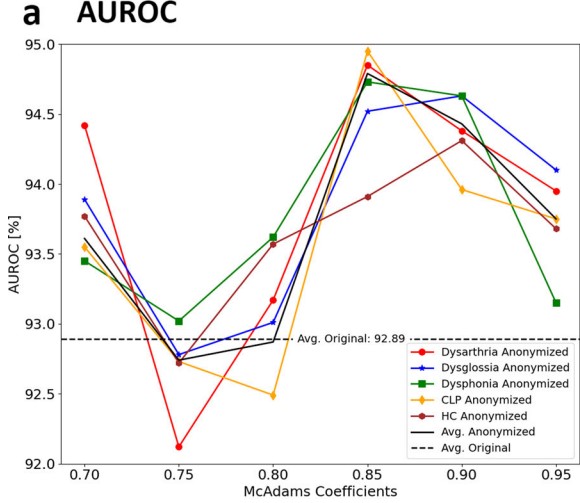

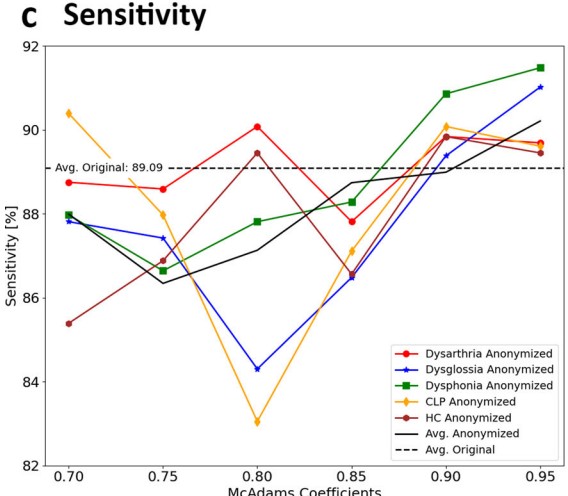

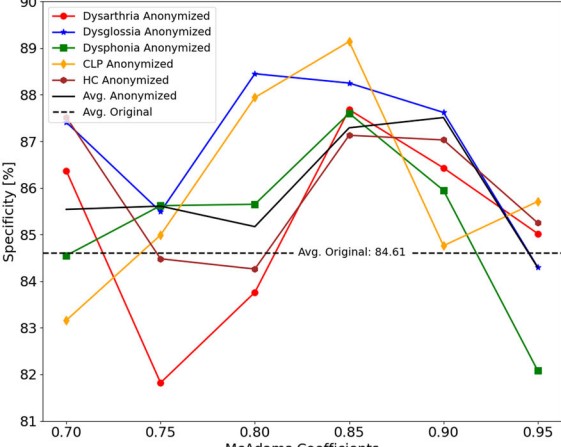

**Fig. 6 | Evaluating the utility of anonymized data for multiclass disorder classification.** The utility of anonymized pathological speech data, anonymized using the McAdams Coefficient method at various privacy levels (as determined by the McAdams coefficient value; see Fig. 3 for details) is illustrated, in a multiclass classification context. This context involves distinguishing among five categories: Healthy Control (HC), Dysarthria, Dysglossia, Dysphonia, and Cleft Lip and Palate (CLP). Utility is measured through (**a**) the area under the receiver operating characteristic curve (AUROC), (**b**) accuracy, (**c**) sensitivity, and (**d**) specificity for each disorder. Colors represent different disorders: red for Dysarthria, blue for Dysglossia, green for Dysphonia, and orange for CLP, with black denoting average values across all disorders. Dotted lines depict the utility curves for the original, non-anonymized speech data. The X-axis charts the varying levels of the McAdams coefficient. The utilized data for this analysis comprised a training set of $n = 372$ speakers and a test set of $n = 96$ speakers. Source data is in Supplementary Data 1.

post-anonymization, a nearly 15-fold increase in difficulty. Although this substantially improves anonymity, manually verifying 30 speakers is still manageable. Therefore, for optimal anonymity enhancement, it is advisable to publish data in larger quantities.

Our study acknowledges certain limitations. First, while our dataset spans various German-speaking regions and exhibits considerable demographic diversity, it is limited to the German language. As part of a validatory experiment, we utilized a PD dataset in Spanish. While some pathological biomarkers were slightly reduced, the results concerning the privacy-utility trade-off were generally consistent with those from our German dataset, indicating the potential generalizability of our proposed methods. However, the results also highlight the urgent need for further research into disorder-specific methods that can effectively address the limitations of automatic anonymization of pathological speech. Additionally, the limited availability of adult controls restricted our ability to completely balance age distributions across all adult sub-groups. Future studies should focus on acquiring more data from both healthy and patient adult populations to deepen and clarify comparative results. Second, our analysis is based on specific speech

tests, suggesting a future exploration of more diverse speech transcripts could offer deeper insights. Third, our primary aim was to assess the utility of anonymization for identifying pathological speech biomarkers, not for general speech recognition which is often evaluated using word recognition or error rates. Regarding privacy, recent studies suggest that the intelligibility of pathological speech, as measured by word recognition rates, does not directly correlate with the ease of speaker identification post-anonymization[13]. On the utility front, our focus was on the utility of anonymized speech in aiding the training of data-driven diagnostic models, where the automatic extraction of features by neural networks is crucial. Fourth, our exploration into inversion methods for anonymized speech was preliminary, a subject that has seen limited discussion in existing studies concerning non-pathological speech. This brief examination demonstrated that such methods could potentially reduce the effectiveness of anonymization, suggesting anonymized speech's vulnerability to re-identification efforts. While some research[72] advocates for the potential reversibility of anonymized speech for trusted entities, concerns arise regarding its compatibility with stringent privacy standards like General Data Protection

Regulation (GDPR)[73] when accessible to untrusted parties[66]. Our future work will delve into these inverse methodologies and their implications for the anonymization of pathological speech. Fifth, while we used disorder detection as the baseline for evaluating the utility, we recognize that this approach may be general for assessing the complex impacts on the data. As such, future research should focus on the degree to which anonymized samples can be investigated with respect to detailed speech analytic measures, including prosodic, phonetic, phonation, and resonance features. This would offer a more granular understanding of the trade-offs involved in anonymization and its effects on the diagnostic quality of speech data. Additionally, we plan to conduct a thorough perceptual analysis of anonymized speech to evaluate its utility not only for machine-based analyses but also for human assessments, thus broadening the scope of utility evaluation. This investigation represents an initial step into this area, with further research necessary to fully understand the impacts on dialects, age, gender, and other variables.

In sum, our study demonstrates that anonymization can substantially increase patient privacy in pathological speech data without substantially compromising diagnostic utility or fairness, marking a pivotal step forward in the responsible use of speech data in healthcare. Further research is needed to refine these methods and explore their application across a broader range of disorders and languages, ensuring global applicability, fairness, and robustness against inversion attacks.

## Data availability
The German speech dataset used in this study is internal data from patients at the University Hospital Erlangen and is not publicly available due to patient privacy regulations. Access requests can be directed to the corresponding author for on-site access at the University Hospital Erlangen in Erlangen, Germany. The PC- GITA[67] dataset is a restricted-access resource. To gain access, users must agree to the dataset's data protection requirements by submitting a request and signing a user agreement through the GITA Lab, University of Antioquia, Medellín, Colombia (rafael.orozco@udea.edu.co). The source data for Figs. 3, 4, 5 and 6 is available as Supplementary Data 1.

## Code availability
To ensure transparency and facilitate further research, our entire source code is publicly accessible at https://doi.org/10.5281/zenodo.12806213[74]. This repository includes comprehensive details on training protocols, evaluation procedures, data preprocessing, and anonymization processes, promoting reproducibility within the research community. The codebase is implemented in Python v3.9 and employs the PyTorch v1.13 framework for all deep learning tasks.

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

## Acknowledgements
We acknowledge financial support by Deutsche Forschungsgemeinschaft (DFG) and Friedrich-Alexander-Universität Erlangen-Nürnberg within the funding program Open Access Publication Funding. This study was funded by Friedrich-Alexander-Universität Erlangen-Nürnberg, Medical Valley e.V., and Siemens Healthineers AG within the framework of d.hip campus.

## Author contributions
The formal analysis was conducted by S.T.A., A.M., and S.H.Y. and the original draft was written by S.T.A. and corrected by M.S., E.N., A.M., and S.H.Y. The software was developed by S.T.A.; S.T.A., T.A.V., P.A.P.T., T.W., K.P., E.N., A.M., and S.H.Y. provided technical expertise; M.S. provided clinical expertise. The experiments were performed by S.T.A. Statistical analysis was performed by S.T.A. Datasets were provided by M.S., A.M., and S.H.Y.; S.T.A. and T.W. downloaded the datasets from the database. S.T.A. cleaned, organized, and pre-processed the data. E.N., A.M., and S.H.Y. supported the conception of the study and the experiments. S.T.A., A.M., and S.H.Y. designed the study. All authors read the manuscript, contributed to the editing, and agreed to the submission of this paper.

## Funding

## Competing interests
The authors declare no competing interests.
