## [Peer Review File · Communications Medicine]

Reviewers' comments:

Reviewer #1 (Remarks to the Author):

This paper investigates the impact of anonymization techniques on pathological speech privacy and diagnostic utility, using a large dataset of over 2,700 speakers with various speech disorders. It demonstrates that disorder-specific anonymization strategies can enhance privacy with minimal diagnostic utility loss. The study's strengths include its relevance, methodological rigor, and practical implications, though it could benefit from exploring more anonymization methods, evaluating the proposed methods in other databased and addressing potential data biases. Below are summarized my main concerns:

1. Reproducibility and generalization. The study uses a dataset specific to German institutions and speakers. While this provides a robust dataset for analysis, it raises questions about the generalizability of the findings to other languages and datasets. Including a discussion or preliminary results on other datasets and languages could enhance the manuscript's impact.

2. Dataset. Despite efforts to balance the dataset, the dataset used for the experiments is highly imbalanced and this raises concerns about the generalizability of the results and whether the deep learning methods are effectively identifying speech biomarkers for pathological speech classification or just reproducing the imbalance on the training dataset. To give a concrete example, adult participants in the control group are significantly younger (average age 23.98 ± 16.04) than those in the experimental groups (average age ~60 years). This worries me, as it could easily happen that the pathological speech detection system is inferring the patient's age from her/his speech to perform the classification into control/pathological speech. Also, I don't know what is the clinical utility of developing an automatic system for identifying CLP speech when this speech impediment is clearly visible. Please elaborate on this.

Other minor concerns:

1. Github repo is not working.

2. Pag 7. Whye the ResNET34 model is pretrained on the Imagnet dataset? It wouldn't be better to pretrain it on a speech dataset?

3. It would be nice to make available to the research community some speech samples of pathological speech with/without anonymization.

4. The training-based method proposed is not really a traing-based method, but rather a vocoder-based speech modification technique.

Reviewer #2 (Remarks to the Author):

Thank you for asking me to review this paper reporting a study which evaluated training-based and signal-level modification methods to anonymise pathological speech samples. The McAdams Coefficient signal-level modification method was the superior approach and was able to achieve anonymity in the samples whilst maintaining diagnostic accuracy of samples from people with dysarthria, dysglossia, dysphonia, and cleft lip and palate. The results also showed variation across the different sample types by disorder suggesting the need for a specific anonymization level for each disorder type that offers an optimal privacy-utility balance.

My comments relate to pathological speech rather than on the analysis of the technical reports of anonymity achieved.

Introduction

1. This is strong and makes a good case for the work but on page 4, it starts to describe the methods and results. I would suggest simply outlining the aims and objectives here and moving the other content to the methods and results sections.

Methods –

2. Page 5 - More information on where, when, how and why the samples were collected is needed.

3. Page 5 - A table providing a comprehensive overview of the different features we would expect to see for different clinical groups would be useful in place of paragraph 3. I would imagine that it would be helpful to readers if the features explained. Also speakers who were born with CLP may have voice problems (phonation) as mentioned but also they are quite likely to have problems with resonance (hyper or hypo nasality) which is not mentioned.

Results-

4. The investigators used disorder detection as the gold standard. While this is useful, it is a relatively crude method. Future research should look at the degree to which the anonymised samples can be investigated in terms of speech analytic measures relating to prosodic, phonetic, phonation and resonance features. The authors do acknowledge this but it could be made more prominent in the paper.

Minor points

5. Page 5 – change the word ‘average’ to ‘mean’ in the sentence ‘It featured a median participant age of 17, with an average age of 30 years (\pm 25 years standard deviation), covering ages from 3 to 95 years’

Point-by-Point Responses to the Reviewers' Comments

Title:

The impact of speech anonymization on pathology and its limits

Reference number:

COMMSMED-24-0411-T

Journal:

Communications Medicine

General Reply:

Dear Reviewers,

We express our sincere gratitude for the detailed feedback and constructive comments. The feedback has been invaluable in refining the manuscript, and we are enthusiastic about the enhanced clarity and depth it now offers. All concerns raised have been thoroughly addressed, and we are confident that these revisions have strengthened the research presented. We hope the manuscript is now deemed suitable for publication in *Communications Medicine*. Please find individual responses to each Reviewer's comments below.

For ease of reference, all modifications in the manuscript have been **highlighted in yellow**. **Both the clean version of the revised manuscript and the annotated version with these yellow highlights are provided.**

Best regards,
Soroosh Tayebi Arasteh
(for the authors)

Reply to Reviewer #1

COMMSMED-24-0411-T

Title:

The impact of speech anonymization on pathology and its limits

General Comment: *“This paper investigates the impact of anonymization techniques on pathological speech privacy and diagnostic utility, using a large dataset of over 2,700 speakers with various speech disorders. It demonstrates that disorder-specific anonymization strategies can enhance privacy with minimal diagnostic utility loss. The study's strengths include its relevance, methodological rigor, and practical implications, though it could benefit from exploring more anonymization methods, evaluating the proposed methods in other databased and addressing potential data biases. Below are summarized my main concerns.”*

Authors' Response: We sincerely thank the Reviewer for recognizing the relevance, methodological rigor, and practical implications of our study.

Comment 1: *“Reproducibility and generalization. The study uses a dataset specific to German institutions and speakers. While this provides a robust dataset for analysis, it raises questions about the generalizability of the findings to other languages and datasets. Including a discussion or preliminary results on other datasets and languages could enhance the manuscript's impact.”*

Authors' Response: We thank the Reviewer for the insightful comments on the generalizability of our findings. We indeed acknowledge the limitations inherent to using a dataset specific to German speakers and institutions and have discussed these in the manuscript's limitations section. Moreover, In response to the concerns raised:

1) Our dataset, while specific to the German language, is diverse, spanning more than 20 locations across Germany. Although all speakers are native German speakers, they represent a variety of backgrounds with over 30 different parents' native languages, such as German, English, Spanish, Turkish, Russian, Arabic, etc. [1,2] This diversity enhances the dataset's generalizability.

2) Although our dataset is focused on a single language, it is among the largest pathological speech datasets available, to the best of our knowledge. It exceeds in scope and variability when compared to other public pathological speech datasets, such as:

- The Saarbrücken Voice Database [3], which includes electroglottography data and recordings from approximately 2,000 German-speaking individuals.
- The TORGO database [4], which features recordings from 15 individuals, including children and adults with dysarthria.

- The Voice Bank Corpus [5], which includes 29 dysarthric speakers and provides an accent-specific perspective with its Scottish and Northern Irish English recordings.
- The PC-GITA dataset [6], with recordings from 50 PD and 50 healthy native Spanish speakers.
- UASpeech dataset [7], which includes recordings from 15 speakers with varying degrees of dysarthria and 13 healthy controls.

A primary goal for future work is to apply our automatic anonymization methods to our dataset to facilitate its ethical release to the research community.

3) Regarding generalizability, our methods are not dependent on any language-specific features. The ideal scenario for a speaker verification system is to function independently of the language and speech content, focusing solely on the identity characteristics of the speaker. We note that while there may be a preference for spectral domain methods, our results demonstrate the superiority of a signal-processing based approach, specifically the McAdams Coefficient method. The VoicePrivacy Challenge [8-10], a key framework for evaluating automatic anonymization methods, has consistently included the McAdams method as a baseline since its inception in 2020. This method's effectiveness is demonstrated in the challenge's English dataset, establishing it as a baseline in the field and encouraging further research based on these findings.

Authors' Action: In response to the Reviewer's feedback, we have performed a preliminary analysis using the PC-GITA dataset, which includes speakers of Colombian Spanish. To systematically present this work, we have added a new subsection entitled "Generalization of the method beyond German language" within the Methods section. This subsection details the experimental setup and procedures adopted for this analysis, as follows:

"Generalization of the method beyond German language"

The anonymization methods presented are not reliant on language-specific characteristics, demonstrating their adaptability across languages. We validated this generalization using the PC-GITA dataset⁶⁷, which consists of speech recordings from 50 Parkinson's Disease (PD) patients and 50 matched healthy controls (by age and gender), all native Spanish speakers from Colombia. The recordings were collected in accordance with a protocol designed to meet technical specifications and recommendations from experts in linguistics, phoniatry, and neurology. Further details on the dataset are available in the original publication⁶⁷.

For anonymization, we employed the McAdams Coefficient method, similar to that used with the German dataset. We utilized phonemic place of articulation features to distinguish between PD patients and healthy controls. A linear support vector regression machine⁶⁸ was applied to predict the maximum phonation duration. The utility of the method was quantitatively assessed using Pearson's r correlation coefficient, comparing the PD patients and healthy controls."

The corresponding results are included as “Generalization to other languages” section in the **Results**, as follows:

“Generalization to other languages

Supplementary Fig. 1 presents the results of both utility and privacy assessments using the PC-GITA dataset⁶⁷, which includes Spanish-speaking PD patients. The overall EER of the anonymized PD dataset is 34.00%, comparable to the results from the German dataset, where the EERs for Dysarthria, Dysglossia, Dysphonia, and CLP are 36.59%, 34.26%, 38.86%, and 32.19%, respectively. Similarly, a logarithmic decline in EER values is observed with linear increases in the McAdams coefficient, akin to the German dataset. Notably, the magnitude of EER changes increases with further adjustments to the coefficient, underscoring the necessity for disorder-specific configurations in anonymization methods. Like in the German dataset, the anonymization process uniformly conceals the identities of both patients and healthy controls. Initially, the EER for controls is 2.00%, while for PD it is 3.78% (nearly twice as high). After anonymization using a random coefficient, the EER for controls and PD equalizes at 34.9% and 34%, respectively, demonstrating that the initial disparity in identification difficulty is effectively neutralized.

Regarding utility, the original data exhibited a correlation coefficient of 0.71. Post-anonymization, this value decreases to 0.42 with a randomized coefficient and to 0.57 with a fixed coefficient set at 0.8. These findings indicate a reduction in some pathological biomarkers, yet the results align well with those from the German dataset. This alignment supports the generalizability of the proposed methods, while also highlighting the critical need for continued research into disorder-specific anonymization techniques.”

Additionally, we have updated the limitations section in the **Discussion** to reflect new insights. It now reads:

“Our study acknowledges certain limitations. First, while our dataset spans various German-speaking regions and exhibits considerable demographic diversity, it is limited to the German language. As part of a validity experiment, we utilized a PD dataset in Spanish. While some pathological biomarkers were slightly reduced, the results concerning the privacy-utility trade-off were generally consistent with those from our German dataset, indicating the potential generalizability of our proposed methods. However, the results also highlight the urgent need for further research into disorder-specific methods that can effectively address the limitations of automatic anonymization of pathological speech. Moreover, further comprehensive examination is necessary, especially with languages that are substantially different from German, such as Mandarin Chinese, to fully validate our findings. [...]”

Comment 2: *“Dataset. Despite efforts to balance the dataset, the dataset used for the experiments is highly imbalanced and this raises concerns about the generalizability of the results and whether the deep learning methods are effectively identifying speech biomarkers for pathological speech classification or just reproducing the imbalance on the training dataset. To give a concrete example, adult participants in the control group are significantly younger (average age 23.98 ± 16.04) than those in the experimental groups (average age ~ 60 years). This worries me, as it could easily happen that the pathological speech detection system is inferring the patient's age from her/his speech to perform the classification into control/pathological speech. Also, I don't know what is the clinical utility of developing an automatic system for identifying CLP speech when this speech impediment is clearly visible. Please elaborate on this.”*

Authors' Response and Action: The authors thank the Reviewer for their input. We agree with the concern raised regarding dataset balance for the age discrepancy between the adult control and pathological groups, which may potentially bias the pathological speech detection system toward age-related differences in some experiments for adults. While the children subset is balanced, the adult controls are younger on average than the patient groups, as mentioned. Despite our best efforts to balance the dataset through the exclusion criteria described in the **Methods** section, this limitation persists for adult controls due to challenges in accessing more comprehensive adult speech data.

However, it is important to note that we conducted various experiments involving different groups, pathologies, and clinical tasks. Not all subgroups exhibited unbalanced age distribution, and our findings were consistent across these different setups, underscoring that the anonymization system must be tailored to be disorder-specific. This was further validated by the addition of a new Spanish dataset, which was completely balanced in terms of age.

Furthermore, in our initial study [1], where we identified the general vulnerability of pathological speech to reidentification attacks, we systematically balanced all different factors such as gender, age, recording environment, microphone type, and test type, to focus solely on the effect of pathology. We used the same dataset in this study, with even more stringent exclusion criteria to balance it further. Although the objectives of this study differ, they build upon the foundational finding that the general vulnerability of pathological speech to reidentification attacks is an effect independent of external factors such as age. We fully acknowledge the need for future research to validate these findings using more balanced datasets for adults. This necessity aligns with a previously noted limitation (Comment #1) regarding the generalization of our findings to languages beyond German. Consequently, we have updated the limitations section of the **Discussion** as follows:

“Our study acknowledges certain limitations. First, [...] Additionally, the limited availability of adult controls restricted our ability to completely balance age distributions across all adult

sub-groups. Future studies should focus on acquiring more data from both healthy and patient adult populations to deepen and clarify comparative results. Second, [...]”

Regarding the Reviewer’s concern about clinical utility of developing an automatic system for identifying CLP speech, the authors acknowledge the comment. While it is true that identifying CLP from speech characteristics might seem overt, this observation underscores the methodology and objectives of our study.

The primary aim of our research is to demonstrate that, after anonymization, pathological speech data can retain their clinical utility and specific biomarkers to a satisfactory degree. We conducted systematic experiments across a spectrum of clinical tasks varying substantially in complexity, from the relatively straightforward task of CLP detection to challenges like general speech and voice disorder detection through Dysarthria, Dysglossia, or Dysphonia detection, multi-class detection of underlying disorders, and the particularly challenging task of differentiating between PD patients and controls based on mFDA.

Our findings consistently indicate that automatic anonymization can be effectively utilized for pathological speech data. However, the utility of anonymized data varies by disorder, which suggests that disorder-specific configurations and methods need to be tailored for optimal performance. For instance, in the case of CLP detection, which is a relatively direct task, we observed no significant performance decrease ($p=0.139$) post-anonymization. This result implies that although some pathological biomarkers of the speech signal are removed during anonymization, the fundamental characteristics required for CLP detection remain sufficiently intact, making the anonymized speech still useful for this purpose.

Conversely, for the more demanding task of differentiating PD patients from controls, our study revealed that the randomized McAdams method removed more critical biomarkers than the McAdams method with a coefficient of 0.8. This outcome highlights that the preservation of biomarker information in the speech signal is crucial for such complex tasks, emphasizing the importance of choosing the appropriate anonymization configuration based on the specific pathology.

These findings and their implications are emphasized throughout our manuscript, from the **Abstract** to the **Results** and **Discussion** sections, and even in the paper's title, which includes the phrase "and its limits" to signify the complex capabilities and boundaries of current anonymization techniques.

Minor Comment 1: “*Github repo is not working.*”

Authors’ Response and Action: The authors thank the Reviewer for bringing this issue to their attention and apologize for the inconvenience. The GitHub repository was set to private. It has now been made public to facilitate broader access for the research community. Additionally, in line with journal guidelines, the final version of the source code

will be permanently archived on Zenodo and cited in the “**Code availability**” section upon acceptance of the paper and before publication.

Minor Comment 2: *“Pag 7. Whye the ResNET34 model is pretrained on the Imagnet dataset? It wouldn't be better to pretrain it on a speech dataset?”*

Authors' Response:

The authors appreciate the Reviewer's query regarding the choice of pretraining the ResNet34 model on the ImageNet dataset instead of a speech-specific dataset. While it is indeed advantageous to use a domain-specific dataset for pretraining, the use of ImageNet provides significant benefits due to its vast scale and diversity. The mel-spectrogram features, being two-dimensional (80x180), allow for their interpretation as natural images, facilitating the use of advanced image recognition architectures developed for the ImageNet challenge [11-13]. ImageNet's extensive annotated dataset has become a gold standard in the field, providing robust pretraining for convolutional architectures like ResNet, which are optimized for feature extraction from complex visual data [14-17]. Furthermore, as detailed in the **Discussion** section, the focus of our study was on the utility of anonymized speech to aid the training of data-driven diagnostic models, where automatic feature extraction by neural networks is crucial. Although a speech dataset comparable in size to ImageNet would be ideal, currently, to the best of our knowledge, no such dataset exists, limiting the feasibility of this approach.

Authors' Action: In response to the Reviewer's comment, the authors have updated the **Methods** section to clarify the rationale behind the selection of the pretrained model. The revised section now reads:

“Classification process

[...]

Network architecture: Due to the two-dimensional nature of log-Mel-spectrograms, we leveraged state-of-the-art pretrained convolutional networks designed for image classification to maximize feature extraction accuracy^{51,52}. We specifically selected a ResNet34⁵³ model pretrained on the large-scale ImageNet⁵⁴ dataset, which contains over 14 million images across 1,000 categories, [...]

Minor Comment 3: *“It would be nice to make available to the research community some speech samples of pathological speech with/without anonymization.”*

Authors' Response and Action: The authors appreciate the Reviewer's suggestion regarding the availability of anonymized pathological speech samples. While we fully

understand the importance of making such data accessible to foster further research, there are stringent patient privacy constraints that prevent the public sharing of patient data. However, as detailed in the "**Data availability**" section, researchers can request access to the data by contacting the corresponding author to arrange on-site data review at the University Hospital Erlangen in Erlangen, Germany.

One of our primary objectives from this study is to demonstrate the potential of automatic anonymization methods to call for more research in this domain and to move closer to being able to publicly release these valuable datasets to the research community without compromising patient privacy. This motivation is emphasized in the first paragraph of the **Introduction**, where we state:

"[...] Therefore, finding ways to expand the pool of publicly available training data without breaching privacy norms is crucial for the progression of medical speech technology applications."

Additionally, as part of our ongoing research efforts, we plan to conduct a comprehensive perceptual study of the anonymized speech data. This is discussed in the **Discussion** section of our manuscript:

"[...] Additionally, we plan to conduct a thorough perceptual analysis of anonymized speech to evaluate its utility not only for machine-based analyses but also for human assessments, thus broadening the scope of utility evaluation. This investigation represents an initial step into this area, with further research necessary to fully understand the impacts on dialects, age, gender, and other variables."

Minor Comment 4: *"The training-based method proposed is not really a training-based method, but rather a vocoder-based speech modification technique."*

Authors' Response: The authors thank the Reviewer for their critical assessment. We acknowledge the clarification regarding the nature of our method. Indeed, the term "training-based" might misleadingly suggest that the method involves training from scratch. In fact, we utilized a well-established pretrained vocoder for speech synthesis. This does not entail training within the context of our study but rather the application of pretrained models.

Authors' Action: In light of this observation, we have revised the terminology used in our manuscript to more accurately reflect the methodology employed. We have renamed this category of anonymization methods to "deep-learning-based" (abbreviated as DL-based) methods throughout the manuscript to avoid any confusion regarding the training aspect of the vocoder.

Reply to Reviewer #2
COMMSMED-24-0411-T

Title:

The impact of speech anonymization on pathology and its limits

General Comment: *“Thank you for asking me to review this paper reporting a study which evaluated training-based and signal-level modification methods to anonymise pathological speech samples. The McAdams Coefficient signal-level modification method was the superior approach and was able to achieve anonymity in the samples whilst maintaining diagnostic accuracy of samples from people with dysarthria, dysglossia, dysphonia, and cleft lip and palate. The results also showed variation across the different sample types by disorder suggesting the need for a specific anonymization level for each disorder type that offers an optimal privacy-utility balance.*

My comments relate to pathological speech rather than on the analysis of the technical reports of anonymity achieved.”

Authors’ Response: We sincerely thank the Reviewer for their thoughtful analysis and comments.

Comment 1: *“Introduction*

This is strong and makes a good case for the work but on page 4, it starts to describe the methods and results. I would suggest simply outlining the aims and objectives here and moving the other content to the methods and results sections.”

Authors’ Response and Action: We thank the Reviewer for their constructive suggestion. We agree that the introduction should primarily focus on outlining the aims and objectives of the study, rather than delving into methods and results. Accordingly, we have revised the previous 5th and 6th paragraphs of the **Introduction** and relocated the detailed descriptions to their appropriate sections.

The revised part of the **Methods** section reads as follows:

“Anonymization measure (privacy)

To evaluate the effectiveness of anonymization, we employ automatic speaker verification (ASV) techniques, aligning with established standards in the field, such as those set by the VoicePrivacy 2020 and 2022 challenges^{10,14,15}. Utilizing a pretrained module, optimized on the LibriSpeech³⁷ dataset, we fine-tune this system to recognize specific speakers. This involves exposing the module to random utterances, which may belong to the target speaker or an imposter from the dataset. The goal is to achieve an equal error rate (EER) where the rate of false acceptance (FA) matches that of false rejection (FR), indicating the

system's difficulty in distinguishing the speaker's identity post-anonymization. We adopt the EER as our primary metric for assessing anonymization³⁸, a critical metric in ASV³⁹. [...]"

The revised part of the **Introduction** section reads as follows:

"In response, our study conducts a comprehensive analysis of the impact of anonymization on pathological speech biomarkers, utilizing a large-scale dataset of over 2,700 speakers from various institutions. This dataset includes a wide array of disorders such as Dysarthria²⁸ (a motor speech disorder affecting muscle control), Dysglossia²⁹ (a condition affecting speech by changes of the articulatory organs, e.g., due to oral cancer), Dysphonia³⁰ (voice disorders affecting vocal fold vibration), and Cleft Lip and Palate (CLP)³¹⁻³⁴ (a congenital split in the upper lip and roof of the mouth), thus providing a broad basis for generalizable insights into pathological speech anonymization. Additionally, we meticulously explore the balance between privacy enhancement and the utility of pathological speech data, including an examination of demographic fairness implications.

This study aims to investigate whether anonymization modifies the diagnostic markers within pathological speech while balancing privacy-utility and privacy-fairness considerations, suggesting the potential for applying automatic anonymization to pathological speech. Additionally, we aim to explore the variability in the effects of anonymization across different disorders, highlighting the complex interactions between anonymization processes and the specific characteristics of pathological conditions."

Comment 2: *"Methods –*

Page 5 - More information on where, when, how and why the samples were collected is needed."

Authors' Response and Action: We thank the Reviewer for highlighting the need for more detailed information regarding the collection and characteristics of our speech dataset. In response, we have substantially revised the **Methods** section of our manuscript to include a comprehensive description of the dataset, data collection procedures, and the ethical considerations adhered to during the study.

Revised **Methods** section reads as follows:

"Ethics statement

The study and the methods were performed in accordance with relevant guidelines and regulations and approved by the University Hospital Erlangen's institutional review board with application number 3473 and respected the Declaration of Helsinki. Informed consent was obtained from all adult participants as well as from parents or legal guardians of the children."

“Speech Dataset

The dataset used in our research comprised a wide array of speech utterances from across Germany. It featured a median participant age of 17, with a mean age of 30 years (± 25 years standard deviation), covering ages from 3 to 95 years. Table 1 offers an overview of the dataset demographics, including voice and speech disorder distributions, and gender breakdown.

Data collection

Data were collected from 2006 to 2019 during regular outpatient examinations at the University Hospital Erlangen and at over 20 different locations across Germany for the recording of control speakers. Every patient during a specialized consultation was invited to participate in the study. Patients and control speakers were informed about the study's procedure and goals before consenting to participate. Recordings were made using a standardized procedure which included consistent settings, microphone setups, and speech tasks. Non-native speakers and patients whose speech was significantly disturbed by factors other than the targeted disorders were excluded. The Program for Evaluation and Analysis of all Kinds of Speech disorders (PEAKS)³⁵, an open-source tool widely used in the German-speaking scientific community, was employed to document and manage the database. Recordings were captured at a 16 kHz sampling frequency and a 16-bit resolution, featuring subjects who are native German speakers, including various local dialects.

Speech features

The dataset included different causes with their main or prominent features of pathologic speech, e.g., “Dysphonia”, refers to voice disorder containing phonation features, “Dysglossia” refers to articulation disorders containing mostly phonetic and sometimes phonation features, “Dysarthria” refers to speech disorder containing phonation, phonetic and prosody features, and “CLP” refers to speech and resonance disorder containing phonetic features, hyper- and hyponasality, and sometimes phonation features. Supplementary Table 1 provides an overview of the expected features for different clinical groups.

Exclusion criteria

The cohort employed in our study represents a meticulously curated subset of the dataset described in¹³, where it is delineated that our initial collection consisted of 216.88 hours of recordings from $n=4,121$ subjects. To refine this dataset to a clean and unbiased selection, we adhered to all exclusion criteria mentioned in¹³, which encompassed data cleaning, ensuring speech quality and noise standards, and the elimination of multi-speaker utterances. Additional steps undertaken in this study include: i) Acknowledging the distinct speech characteristics between adults and children¹³, we categorized the dataset into two

primary subsets. Adults, defined as individuals over 20 years of age, were tasked with reading "Der Nordwind und die Sonne", a phonetically rich German adaptation of Aesop's fable "The North Wind and the Sun".³⁵ This text comprises 108 words, 71 of which are unique. Conversely, children participated in the "Psycholinguistische Analyse kindlicher Sprechstörungen" (PLAKSS)³⁶ test, which involved naming pictograms across slides, covering all German phonemes in various positions. Given the tendency of some children to describe pictograms with multiple words, and the occasional extra words between target words, recordings were automatically segmented at pauses exceeding 1s³⁵. ii) Adults' subset focused on utterances characterized by Dysarthria²⁸, Dysglossia²⁹, and Dysphonia³⁰, alongside healthy control samples. Utterances with ambiguous or mixed pathologies or those representing conditions with a scant number of data points were excluded. iii) For children, the emphasis was placed on utterances from individuals with CLP conditions — the most prevalent cranial malformation characterized by an incomplete closure of the vocal tract^{31,32,34} — as well as from healthy controls."

Comment 3: "Methods –

Page 5 - A table providing a comprehensive overview of the different features we would expect to see for different clinical groups would be useful in place of paragraph 3. I would imagine that it would be helpful to readers if the features explained. Also speakers who were born with CLP may have voice problems (phonation) as mentioned but also they are quite likely to have problems with resonance (hyper or hypo nasality) which is not mentioned."

Authors' Response and Action: We thank the Reviewer for their constructive suggestions. We acknowledge the need for a more detailed presentation of the speech characteristics associated with different clinical groups to enhance the manuscript's clarity and utility to readers.

In response to the suggestion, we have created a new table (Supplementary Table 1) that provides an overview of the different speech features typically associated with clinical conditions including Dysarthria, Dysglossia, Dysphonia, and CLP. Due to the journal's limitation of up to 10 display items in the main manuscript, and given that we have already reached this limit, we have included this new table as a supplementary item. We believe this addition will make the information more accessible and beneficial for readers. Additionally, the Reviewer's point regarding hyper or hypo nasality, is well-taken and has been incorporated into the revised manuscript.

To facilitate easier navigation and understanding of the speech features, which the Reviewer correctly identified as crucial, we have revised the dataset section of the **Methods** to include a new subsection titled "**Speech features.**"

Revised "**Speech features**" subsection in the **Methods**:

“Speech features

The dataset included different causes with their main or prominent features of pathologic speech, e.g., “Dysphonia”, refers to voice disorder containing phonation features, “Dysglossia” refers to articulation disorders containing mostly phonetic and sometimes phonation features, “Dysarthria” refers to speech disorder containing phonation, phonetic and prosody features, and “CLP” refers to speech and resonance disorder containing phonetic features, hyper- and hyponasality, and sometimes phonation features. Supplementary Table 1 provides an overview of the expected features for different clinical groups.”

Comment 4: “Results –

The investigators used disorder detection as the gold standard. While this is useful, it is a relatively crude method. Future research should look at the degree to which the anonymised samples can be investigated in terms of speech analytic measures relating to prosodic, phonetic, phonation and resonance features. The authors do acknowledge this but it could be made more prominent in the paper.”

Authors’ Response: We thank the Reviewer for their constructive feedback. We fully agree with the observation that our current method of disorder detection, while effective as a baseline, warrants further investigation in future studies.

Our research has demonstrated that after anonymization, pathological speech data can retain their clinical utility and specific biomarkers to a satisfactory degree across a range of clinical tasks, from the relatively straightforward task of CLP detection to more complex challenges such as differentiating between PD patients and controls based on mFDA. While we found that the automatic anonymization of pathological speech data is viable, the effectiveness of anonymization does vary by disorder, indicating a need for disorder-specific configurations for optimal performance. This view has been consistently emphasized throughout our manuscript, from the **Abstract** to the **Results** and **Discussion** sections, and is even reflected in the manuscript’s title, which includes the phrase "and its limits," to signify the complex capabilities and boundaries of current anonymization techniques.

Authors’ Action: In response to the Reviewer’s comments and to emphasize the potential for more detailed speech analysis, we have made revisions to the limitations section of the **Discussion** in our manuscript:

“Our study acknowledges certain limitations. [...] Fifth, while we used disorder detection as the baseline for evaluating the utility, we recognize that this approach may be general for assessing the complex impacts on the data. As such, future research should focus on the degree to which anonymized samples can be investigated with respect to detailed speech

analytic measures, including prosodic, phonetic, phonation, and resonance features. This would offer a more granular understanding of the trade-offs involved in anonymization and its effects on the diagnostic quality of speech data. [...]"

Comment 5: *“Minor points*

Page 5 – change the word ‘average’ to ‘mean’ in the sentence ‘It featured a median participant age of 17, with an average age of 30 years (\pm 25 years standard deviation), covering ages from 3 to 95 years.’”

Authors’ Response and Action: We thank the Reviewer for their suggestion. We have replaced “average” with “mean” as recommended.

References:

- [1] Tayebi Arasteh, S., Weise, T., Schuster, M., Noeth, E., Maier, A., & Yang, S. H. (2023). The effect of speech pathology on automatic speaker verification: a large-scale study. *Scientific Reports*, 13(1), 20476.
- [2] Maier, A. et al. Peaks - A system for the automatic evaluation of voice and speech disorders. *Speech Commun.* 51, 425–437. (2009).
- [3] Barry, W. & Putzer, M. Saarbrücken Voice Database.
- [4] Rudzicz, F., Namasivayam, A. K. & Wolff, T. The TORGO database of acoustic and articulatory speech from speakers with dysarthria. *Lang Resources & Evaluation* 46, 523–541 (2012).
- [5] Veaux, C., Yamagishi, J. & King, S. The voice bank corpus: Design, collection and data analysis of a large regional accent speech database. in 2013 International Conference Oriental COCOSDA held jointly with 2013 Conference on Asian Spoken Language Research and Evaluation (O-COCOSDA/CASLRE) 1–4 (IEEE, 2013).
- [6] Orozco-Aroyave, Juan Rafael, et al. "New Spanish speech corpus database for the analysis of people suffering from Parkinson's disease." LREC. 2014.
- [7] Heejin Kim, Mark Hasegawa Johnson, Jonathan Gunderson, Adrienne Perlman, Thomas Huang, Kenneth Watkin, Simone Frame, Harsh Vardhan Sharma, Xi Zhou, March 17, 2023, "UASpeech", IEEE Dataport.
- [8] Tomashenko, N. et al. Introducing the VoicePrivacy Initiative. in INTERSPEECH 2020 1693–1697 (ISCA, 2020). doi:10.21437/Interspeech.2020-1333.
- [9] Tomashenko, N. et al. The VoicePrivacy 2020 Challenge: Results and findings. *Computer Speech & Language* 74, 101362 (2022).
- [10] Tomashenko, N. et al. The VoicePrivacy 2022 Challenge Evaluation Plan. Preprint at <http://arxiv.org/abs/2203.12468> (2022).
- [11] Elfaki, Ayman, et al. "Using the short-time fourier transform and resnet to diagnose depression from speech data." 2021 IEEE International Conference on Computing (ICOCO). IEEE, 2021.
- [12] M. Muzammel, H. Salam, Y. Hoffmann, M. Chetouani and A. Othmani, AudVowelConsNet: A phoneme-level based deep CNN architecture for clinical depression diagnosis, vol. 2, pp. 100005.
- [13] K. Chlasta, K. Wolk and I. Krejtz, Automated speech-based screening of depression using deep convolutional neural networks, vol. 164, pp. 618-628.
- [14] Krizhevsky A, Sutskever I, Hinton GE (2017) ImageNet classification with deep convolutional neural networks. *Commun ACM* 60:84–90.
- [15] Beyer L, Hénaff OJ, Kolesnikov A, Zhai X, Oord A van den (2020) Are we done with ImageNet? arXiv. <https://doi.org/10.48550/arXiv.2006.07159>.
- [16] Tayebi Arasteh, Soroosh, et al. "Enhancing diagnostic deep learning via self-supervised pretraining on large-scale, unlabeled non-medical images." *European Radiology Experimental* 8.1 (2024): 10.
- [17] Ke A, Ellsworth W, Banerjee O, Ng AY, Rajpurkar P (2021) CheXtransfer: performance and parameter efficiency of ImageNet models for chest X-ray interpretation. In: *Proceedings of the Conference on Health, Inference, and Learning. Virtual Event USA: ACM* 116–124.

REVIEWERS' COMMENTS:

Reviewer #1 (Remarks to the Author):

The authors addressed all my concerns from the previous version in the rebuttal. I think the paper is now ready for publication.

Congratulations to them for this excellent piece of work!

Reviewer #2 (Remarks to the Author):

This interesting paper describes a process for anonymisation of pathological speech samples and also reports the effectiveness of this approach. It is a novel and much needed procedure which has potential for widespread use. While the technological advances which have led to this work will no doubt be further developed and improved upon over time, this report is important in providing a clear description of the process used with the potential for reproducibility.

I am happy that the concerns raised in my original review have been satisfactorily addressed.

Yvonne Wren, Professor of Speech and Communication, University of Bristol

Point-by-Point Responses to the Reviewers' Comments

Title:

Challenges in speaker anonymization in maintaining utility while ensuring privacy of pathologic speech

Reference number:

COMMSMED-24-0411-A

Journal:

Communications Medicine

Reply to Reviewer #1

General Comment: *"The authors addressed all my concerns from the previous version in the rebuttal. I think the paper is now ready for publication.*

Congratulations to them for this excellent piece of work."

Authors' Response: We thank the Reviewer for their positive feedback and are pleased to have their support for publication.

Reply to Reviewer #2

General Comment: *"This interesting paper describes a process for anonymisation of pathological speech samples and also reports the effectiveness of this approach. It is a novel and much needed procedure which has potential for widespread use. While the technological advances which have led to this work will no doubt be further developed and improved upon over time, this report is important in providing a clear description of the process used with the potential for reproducibility.*

I am happy that the concerns raised in my original review have been satisfactorily addressed.

Yvonne Wren, Professor of Speech and Communication, University of Bristol."

Authors' Response: We appreciate Professor Wren's recognition of the manuscript's importance and are grateful for her endorsement of the revisions made.